# Screening and dynamic change study of microbial and metabolite markers for calf diarrhea based on multi-omics and machine learning

Xitong Yin,[1] Yanlong Niu,[1] Baoxia Chen,[1] Hao Zhang,[1] Rongxia Guo,[1] Chun Niu,[1] Jianguo Kang,[1] Hongmei Shi,[2] Xiangying Kong,[3] Weidong Ma,[4] Zhongfa Ma,[5] Yanming Wei,[1] Yongli Hua[1]

**ABSTRACT**   Neonatal calf diarrhea is a leading cause of calf mortality and substantial economic loss in the livestock industry, yet the dynamic microbial and metabolic signatures accompanying disease onset remain poorly defined. Here, we integrated 16S rRNA high-throughput sequencing, untargeted metabolomics, and machine learning approaches to longitudinally profile fecal samples from neonatal calves at 0, 5, 10, 15, and 20 days of age. Diarrheic calves exhibited significantly reduced gut microbial α-diversity, as indicated by lower Chao1 richness and Shannon index compared with healthy counterparts. At the genus level, *Tyzzerella* and *Fusobacterium* emerged as core differential taxa associated with diarrhea and were further validated as robust biomarkers using an XGBoost predictive model. Metabolomic analysis showed that differential metabolites were mainly enriched in pathways including the phosphotransferase system. Notably, dulcitol, N-acetylmuramate, and D-fructose were highlighted as potential contributors to diarrhea, possibly through modulating intestinal osmolality and inflammatory responses. Pearson correlation analysis revealed significant associations between *Tyzzerella/Fusobacterium* and key metabolites, suggesting coordinated microbe–metabolite interactions during disease progression. Temporal pattern analysis identified an early-life signal: a high abundance of *Escherichia–Shigella* at birth may act as an important trigger for subsequent diarrhea. In addition, several metabolites displayed distinct age-dependent trajectories, indicating their potential as time-resolved metabolic biomarkers. Collectively, this study delineates dynamic shifts in the gut microbiome and metabolome during neonatal calf diarrhea, identifies *Tyzzerella* and *Fusobacterium*, together with characteristic metabolites such as dulcitol and N-acetylmuramate, as candidate biomarkers, and provides a high-performance predictive framework to support early diagnosis and targeted microbiota-based interventions.

**IMPORTANCE** Neonatal calf diarrhea causes substantial early-life mortality and economic losses, yet the dynamic microbiota–metabolite alterations and early-warning biomarkers during disease onset remain poorly defined. Here, we longitudinally profiled fecal microbiota and metabolites in calves from birth to day 20 and integrated machine learning approaches to systematically characterize diarrhea-associated signatures. Diarrheic calves showed reduced α-diversity, and *Tyzzerella* and *Fusobacterium* emerged as core differential genera with predictive value validated using an XGBoost model. Differential metabolites were mainly enriched in pathways such as the phosphotransferase system (PTS), and dulcitol and N-acetylmuramate may contribute to diarrhea by modulating intestinal osmolality or inflammatory responses. Notably, a higher abundance of *Escherichia–Shigella* at birth was potentially associated with subsequent diarrhea risk, while L-glutamic acid, choline, and LysoPC exhibited distinct temporal trajectories. Collectively, these findings provide translational candidate

**Peer Reviewer** Yuting Zhai, University of Florida, Gainesville, Florida, USA

Address correspondence to Yongli Hua, huayongli2004@163.com.

The authors declare no conflict of interest.

biomarkers to support early warning and microbiota-targeted precision interventions for neonatal calf diarrhea.

KEYWORDS calf diarrhea, metabolism, machine learning, microorganisms

Calf diarrhea is a leading cause of mortality in newborn calves, with the highest incidence occurring within the first month of life (1, 2). This condition results from complex interactions between pathogenic agents and environmental factors and environmental factors and poses significant economic challenges to the dairy industry worldwide (3). For instance, studies indicate that the incidence of diarrhea among neonatal calves on dairy farms in the southern United States can reach 80%, with a mortality rate of approximately 20% (4). In Japan, the economic losses due to calf diarrhea were estimated at around 1 billion JPY in 2017 (5). In South Korea, diarrhea was reported in 43.6% of dairy calves and 68.7% of beef calves (6). In Canada, the average treatment cost for per diarrheic calf was approximately 36 USD in 2020 (7). As beef and dairy production in China continues to intensify, calf diarrhea has emerged as a major constraint to the sustainable development of the industry. It is important to note that farming management practices and calf health recording systems in China differ considerably from those in other countries (8, 9). Therefore, conducting risk factor analyses and predictive studies adapted to China's specific conditions is crucial for supporting the sustainable growth of the livestock sector.

The rapid development of high-throughput sequencing has highlighted the close association between gut microbiota dysbiosis and disease occurrence (10). Diarrhea significantly reduces microbial richness and diversity, whereas maintaining ecological balance enhances host resistance (11, 12). The Firmicutes-to-Bacteroidetes ratio, a key indicator of gut health, is markedly altered during diarrhea but tends to normalize after recovery (13–15). Complementarily, untargeted metabolomics enables systematic profiling of metabolic alterations and has been applied to identify disease biomarkers, such as chenodeoxycholic acid and creatinine for colorectal cancer, or indole-related metabolites linked to acute diarrhea in dogs (16–19). By integrating high-dimensional multi-omics data, machine learning further advances biomarker discovery, capturing nonlinear associations and enhancing predictive accuracy. Its application in early cancer diagnostics illustrates its potential for translation into clinical and agricultural practices, offering novel opportunities for precision animal health management (20, 21).

This study aims to identify key microbial and metabolic biomarkers associated with calf diarrhea and characterize their dynamic variations across different age groups. Using 16S rRNA high-throughput sequencing and untargeted metabolomics, we systematically compared the microbial community structures and metabolic profiles of fecal samples from diarrheic and healthy calves. By integrating multi-omics data with machine learning algorithms, including random forest and logistic regression, we discerned microbial taxa and metabolites significantly linked to diarrhea and developed a high-precision predictive model. These results are expected to elucidate the microbial–metabolic interaction mechanisms underlying the onset and progression of calf diarrhea, thereby providing a theoretical basis and technical support for early warning and precision intervention strategies.

## MATERIALS AND METHODS

### Experimental animals and time

The experiment was carried out at a dairy farm from September 2024 to January 2025. The experimental animals were all newborn Holstein calves. The birth weight was 39 ± 3 kg.

## Sample collection

After birth, all calves were orally administered 4 L of colostrum and individually housed in calf hutches (2.0 m × 1.5 m). Fecal samples were collected at 0, 5, 10, 15, and 20 days of age by rectal stimulation. During this period, calves were fed heat-sterilized milk and milk replacer, and the calf hutches were disinfected twice daily (morning and evening). Immediately after collection, fecal samples were transferred into sterile cryovials, snap-frozen in liquid nitrogen, and subsequently stored at −80°C until further analysis. A total of 130 calves were initially enrolled. Calves showing clinical signs of disease at birth, those with congenital defects, or those born to dams with a history of infectious disease were excluded. The diagnosis of calf diarrhea is assessed using the Monica Probo method (22). Calf fecal consistency was scored daily throughout the study period. When a calf reached a fecal score of 3–4, it received symptomatic treatment administered by a licensed veterinarian. Because such interventions may alter the gut microbiota, metabolite profiles, and other biological parameters, fecal scoring and sample collection were discontinued for that calf thereafter. Only calves that developed diarrhea at the predefined sampling time points were included in the study and completed sample collection, whereas calves that developed diarrhea outside the scheduled sampling time points were excluded from subsequent analyses. The overall experimental workflow of this study is illustrated in Fig. 1. Of the 130 calves initially enrolled, 80 were ultimately included in the data analysis, consisting of 20 normal calves (with 100 samples), 20 diarrheic calves at 10 days of age (with 60 samples), 20 diarrheic calves at 15 days of age (with 80 samples), and 20 diarrheic calves at 20 days of age (with 100 samples). The incidence of diarrhea in calves is shown in Fig. 2.

## Sample numbering and grouping treatment

Table S1 provides a detailed description of the sample numbering system, its significance, and the sample size. Table S2 outlines the criteria for selecting samples from the total collection to form analytical groups, based on different analytical objectives.

## Analysis of 16SrRNA gene sequences in microorganisms

The samples (*n* = 300) were sent to Beijing Novogene Technology Co., Ltd. for analysis. DNA was extracted using a DNA extraction kit (DP336; TianGen, Beijing, China), and its purity and concentration were assessed using 2% agarose gel electrophoresis and NanoDrop (NC 2000, Thermo Fisher Scientific, Waltham, MA, USA). The bacterial 16S rRNA gene V3–V4 region was PCR amplified using universal primers 341 F (5′-CCTAYGGGRBGCASCAG-3′) and 806 R (5′-GGACTACHVGGGTWTCTAAT-3′). The cycling conditions included an initial denaturation step at 98°C for 1 min, followed by 30 cycles of denaturation at 98°C for 10 s, annealing at 50°C for 30 s, and extension at 72°C for 30 s, with a final extension at 72°C for 5 min. PCR products were electrophoresed on a 2% agarose gel for detection. Qualified PCR products were purified using magnetic beads, and enzyme-based quantification was performed. Equal amounts of PCR products were pooled, mixed thoroughly, and then electrophoresed on a 2% agarose gel. The target bands were excised, and the PCR amplicons were purified. Library construction was performed using the Qubit 2.0 Fluorometer (Thermo Fisher Scientific, USA). The constructed libraries were quantified by Qubit and Q-PCR, and sequencing was carried out on the NovaSeq 6000 platform (Beijing Novogene Technology Co., Ltd.) with PE250 sequencing (23). The sequencing data have been deposited in the NCBI Sequence Read Archive (SRA) under accession number PRJNA1330053.

## Fecal metabolomics analysis

The fecal samples were slowly thawed at 4°C, and 25 mg of each sample was weighed and placed in a 1.5 mL Eppendorf tube. To this, 800 µL of extraction solvent (methanol:acetonitrile:water = 2:2:1, vol:vol:vol, pre-cooled at −20°C) and 10 µL of internal standard were added. Two small steel beads were placed in the tube, and the samples

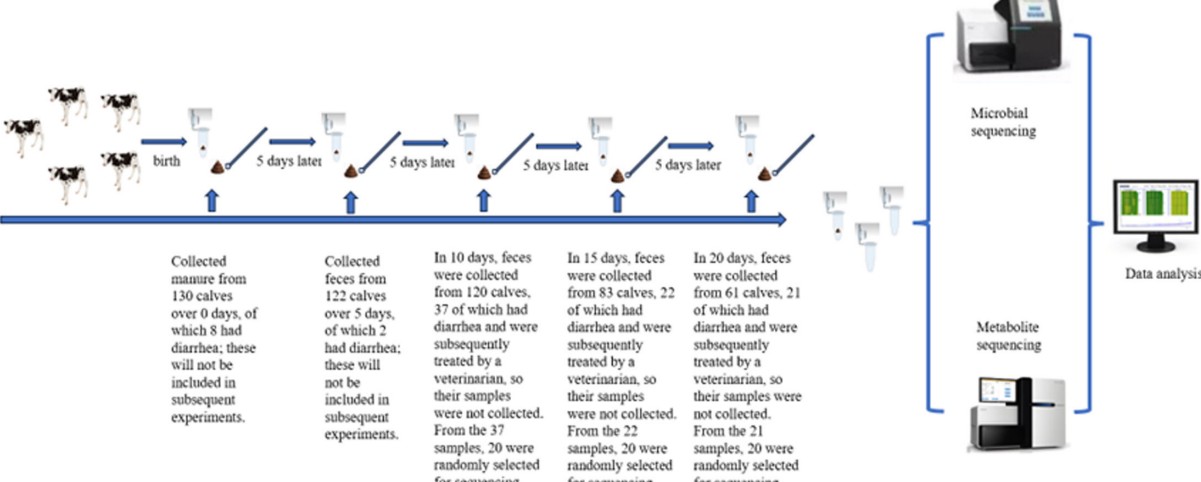

FIG 1 Experimental procedure and sample collection. Samples were collected from neonatal calves every 5 days after birth, stored in cryopreservation tubes, and sequenced after all collections were finished.

were homogenized using a tissue grinder (50 Hz, 5 min). The mixture was then subjected to ultrasonic treatment in a 4°C water bath for 10 min, followed by incubation at −20°C for 1 h. The samples were then centrifuged at 25,000 rpm for 15 min at 4°C. After centrifugation, 600 µL of the supernatant was collected, evaporated to dryness using a freeze-drying vacuum concentrator, and reconstituted in 600 µL of reconstitution solution (methanol:water = 1:9, vol:vol). The reconstituted sample was vortexed for 1 min, sonicated in a 4°C water bath for 10 min, and centrifuged at 25,000 rpm for 15 min at 4°C. The supernatant was then transferred to a sample vial. Metabolite separation and detection were performed using a Vanquish UHPLC system coupled with a Q Exactive HF high-resolution mass spectrometer. The metabolomics data sets generated and analyzed during the current study are available in the OMIX repository of the National Genomics Data Center (NGDC) under accession number OMIX012239.

## Data processing

### Processing of microbial raw data

After removing the barcode and primer sequences, FLASH (Version 1.2.11, http://ccb.jhu.edu/software/FLASH/) was used to assemble the reads for each sample, resulting in the raw tags data (Raw Tags). The reverse primer sequences were then matched and trimmed using Cutadapt software to eliminate any residual sequences, preventing interference with subsequent analyses. The assembled Raw Tags were subjected to rigorous quality filtering using fastp software (Version 0.23.1) to obtain high-quality tags. Chimeric sequences were detected by aligning the filtered tags against the species annotation database (Silva database, https://www.arb-silva.de/ for 16S/18S) and were subsequently removed. The DADA2 module in QIIME2 (Version QIIME2-202202) was used for denoising to obtain the final amplicon sequence variants (ASVs). Species annotation was performed using QIIME2 software with the Silva138.1 database for taxonomic classification (24).

### Processing of original metabolite data

The raw data files (.raw) were imported into CD 3.3 software for processing. A simple selection of parameters, such as retention time and *m/z* ratio, was performed for each metabolite. The peak area was then corrected using the first QC sample to enhance the accuracy of the identification. Subsequently, parameters including mass deviation (5 ppm), signal intensity deviation (30%), minimum signal intensity, and adduct

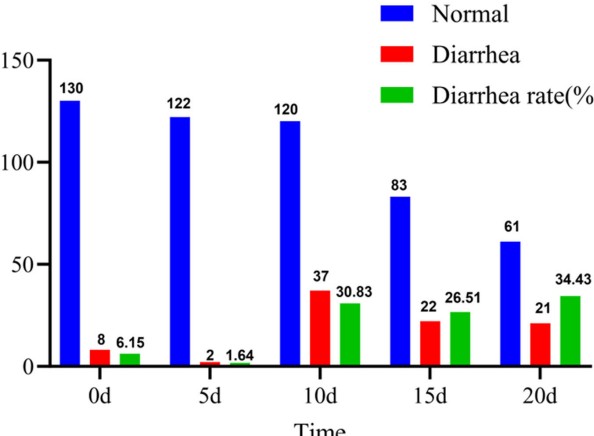

**FIG 2** Number of calves with diarrhea at different sampling times. Normal indicates no diarrhea occurred. Diarrhea indicates diarrhea occurred. When diarrhea occurred, the calves received treatment and were not included in the later trial. After the sampling was completed, 20 samples were selected from each group for the test.

ions were set for peak extraction. Quantification was performed based on the peak area, and target ions were integrated. Molecular formulas were predicted using the molecular ion peaks and fragment ion peaks, and comparisons were made with the mzCloud (https://www.mzcloud.org/), mzVault, and Masslist databases. Blank samples were used to remove background ions. The raw quantification results were normalized using the formula sample raw quantification value/(sample metabolite total quantification value/QC1 sample metabolite total quantification value) to obtain relative peak areas. Compounds with a coefficient of variation (CV) greater than 30% in the relative peak areas in the QC samples were removed. Finally, metabolite identification and relative quantification results were obtained. Data processing was performed on a Linux operating system (CentOS version 6.6) using R and Python software (25).

## Fecal microbiota machine learning

This study employed a systematic workflow to analyze two data sets, including data preprocessing, feature selection, and model construction. During the preprocessing phase, the grouping variable was converted to a factor type, missing values were imputed using random forest, and the data were split into training and validation sets in a 6:4 ratio. Feature selection was performed using the Boruta algorithm and recursive feature elimination, with parameter optimization conducted through cross-validation.

The model construction includes the following:

i.  Random forest (RF): feature selection based on feature importance, optimizing the mtry parameter.
ii.  Support vector machine (SVM): using a radial basis function kernel, with grid search to optimize the penalty coefficient C and kernel parameter gamma.
iii.  eXtreme gradient boosting (XGBoost): feature contribution interpretation using SHAP values, evaluation of multidimensional metrics, and plotting of ROC curves.
iv.  Logistic regression (LR): detection of multicollinearity through the kappa value and outputting interpretable coefficient estimates.
v.  LASSO regression (LR): automatic feature selection using L1 regularization to address high-dimensional data issues.
vi.  Neural Network (NN): built using the Keras framework, employing the Adam optimizer and cross-entropy loss function to capture nonlinear feature interactions.

Multidimensional metrics, including accuracy, sensitivity, specificity, and F1 score, are calculated using the confusion matrix, and overall discriminative ability is assessed through ROC curve analysis (AUC value). The optimal classification threshold is determined using the Youden index: coords (roc_curve, x="best"), visualizing the ROC curve and labeling the AUC value and optimal threshold point. A customized confusion matrix plot (draw_confusion_matrix function) is used to integrate the main evaluation metrics.

## Data analysis

Statistical analysis was performed using SPSS 26.0 software. The Wilcoxon rank-sum test was used for comparisons between two groups, while the Kruskal-Wallis rank-sum test was used for comparisons among multiple groups. A $P$ value <0.05 was considered statistically significant.

## RESULTS

### Comparative analysis based on NC_Group and CD_Group microorganisms

Analysis of the microbiome in the NC_Group (normal group, $n = 120$) and CD_Group (diarrhea group, $n = 60$) revealed significant differences in the microbial community structure and composition. The normal group exhibited a significantly higher number of unique species (2,475 species) compared to the diarrhea group (798 species) (Fig. 3a). α-diversity analysis indices, including Chao1, observed species, and Shannon, were significantly higher in the normal group than in the diarrhea group (Fig. 3b). The β-diversity analysis based on weighted and unweighted UniFrac distances revealed that there was overlap between the normal group and the diarrhea group, and no significant separation was observed in the initial PCoA plot, indicating that the structural profiles of the intestinal microbial communities in the two groups were largely similar. (Fig. S1a). At the phylum level, Firmicutes and Bacteroidota were the dominant phyla in both groups (Fig. S1b). However, the relative abundance of Bacteroidota and Verrucomicrobiota was significantly higher in the normal group, while Fusobacteriota was significantly more abundant in the diarrhea group (Fig. 3c). At the genus level, the dominant genera were *Faecalibacterium*, *Bacteroides*, and *Lactobacillus* (Fig. S1c). Between-group comparisons showed that the relative abundance of *Bacteroides*, *Faecalibacterium*, and *Subdoligranulum* was significantly higher in the normal group, whereas the relative abundance of *Fusobacterium* was significantly higher in the diarrhea group (Fig. 3d). To identify biomarker taxa, LefSe analysis (LDA score = 4) was employed, which indicated that *Faecalibacterium*, *Bacteroides*, *Subdoligranulum,* and *Akkermansia* were characteristic of the normal group, while *Tyzzerella* and *Fusobacterium* were associated with the diarrhea group (Fig. 3e).

### Metabolite analysis based on NC_Group and CD_Group

In both positive and negative ion modes, the base peak chromatograms (BPC) of all samples overlapped well, indicating high instrument stability throughout the analysis and confirming that the data quality meets the requirements for subsequent analysis. In the PCA score plot of the total samples, the quality control (QC) samples were tightly clustered, demonstrating the high reproducibility and reliability of the experiment (Fig. 4a). Further PCA analysis revealed that the samples within the normal group and diarrhea group clustered separately, showing good parallelism. Additionally, a distinct separation between the two groups was observed, indicating a comprehensive metabolic difference between the normal group and the diarrhea group (Fig. 4b). To maximize the inter-group differences and identify potential biomarkers, supervised partial least squares discriminant analysis (PLS-DA) was applied. The constructed PLS-DA model effectively distinguished between the normal group and diarrhea group, with model quality meeting the analytical requirements (Fig. 4c). In both positive and negative ion modes, the fecal metabolites of the two groups exhibited significant separation on the PLS-DA score plot

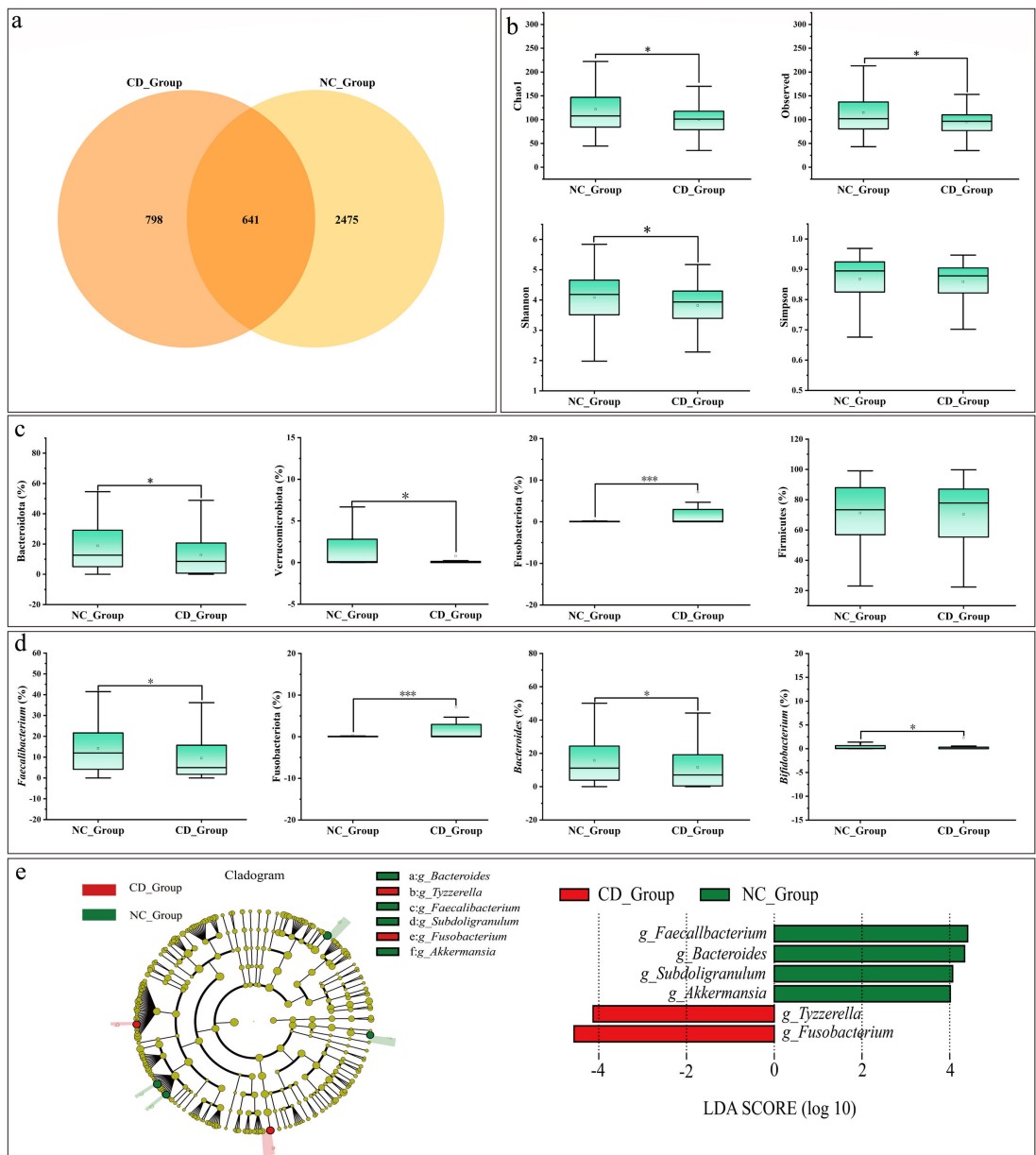

**FIG 3** Fecal microbiota analysis in normal and diarrheal conditions. (a) Venn diagram of OTU distribution between NC_Group and CD_Group. (b) α-Diversity analysis. (c) Phylum-level taxon diversity. (d) Genus-level taxon diversity. (e) LefSe analysis. The evolutionary branching diagram is shown on the left. The LDA score graph is shown on the right. The asterisks denote significant *P* values: \*P < 0.05; \*\*P < 0.01; \*\*\*P < 0.001. Box plots show the median, upper and lower quartiles, and whiskers correspond to 1.5 times the interquartile range.

(Fig. 4d), further confirming their distinct metabolic states. Based on the PLS-DA model, the validity of the model was confirmed through permutation testing, and differential metabolites were selected according to the criteria of variable importance in projection (VIP > 1.0), fold change (FC > 1.2 or FC < 0.833), and statistical significance (P < 0.05). The results showed that in the positive ion mode, a total of 510 differential metabolites were identified, with 464 metabolites upregulated and 46 metabolites downregulated (Fig. 4e). In the negative ion mode, a total of 308 differential metabolites were identified, with 294 metabolites upregulated and 14 metabolites downregulated (Fig. 4f). KEGG pathway enrichment analysis was conducted on these differential metabolites to reveal their potential biological functions. In the positive ion mode, the differential metabolites were significantly enriched in 20 pathways (Fig. 4g). Key differential metabolites involved

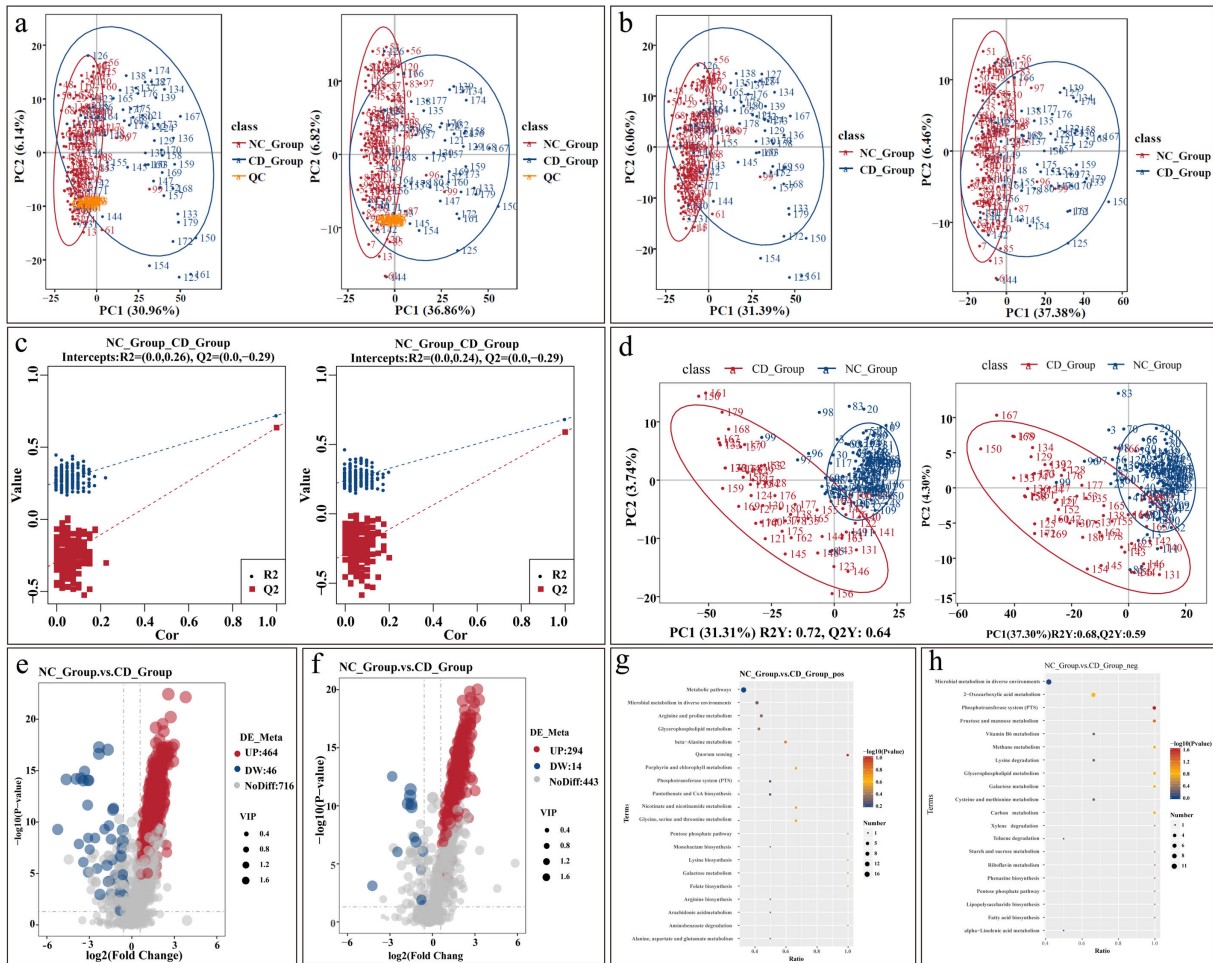

**FIG 4** Normal and diarrhea metabolome analysis (the left panel is the positive ion mode, and the right panel is the negative ion mode). (a) Total sample PCA analysis. (b) Sample PCA analysis. (c) PLS-DA displacement test result graph. (d) PLS-DA score plot. (e) Volcano plot of differential metabolites in the positive ion mode. (f) Volcano plot of differential metabolites in the negative ion mode. (g) Enrichment analysis of differentially expressed metabolites in the KEGG pathway under the positive ion mode. (h) KEGG pathway enrichment analysis of differential metabolites in the negative ion mode.

in these pathways include N-3-hydroxyoctanoyl-L-homoserine lactone, 4-aminobutyric acid, dihydrouracil, and vitamin B5. In the negative ion mode, the differential metabolites were also significantly enriched in 20 pathways (Fig. 4h). Key differential metabolites in this mode included dulcitol, 2-O-(α-D-mannosyl)-D-glycerate, N-acetylmuramate, and D-fructose. Notably, the PTS pathway exhibited significant differences between the two groups, suggesting that bacterial sugar uptake mechanisms may undergo important alterations during the course of diarrhea.

## Correlation analysis based on NC_Group and CD_Group

To investigate the associations between gut microbiota and metabolites, we performed Pearson correlation analysis on differential microbial genera and differential metabolites that were significantly enriched in key metabolic pathways. The results revealed that in negative ion mode, N-acetylmuramate exhibited a highly significant positive correlation with *Bacteroides* and a highly significant negative correlation with *Fusobacterium* and *Tyzzerella*. D-fructose showed a significant negative correlation with *Fusobacterium* and *Tyzzerella* and a highly significant positive correlation with *Escherichia–Shigella*. Dulcitol demonstrated a highly significant positive correlation with *Prevotella_7*. 2-O-(α-D-mannosyl)-D-glycerate was positively correlated with *Prevotella_7* and *Escherichia–Shigella* at

a highly significant level, while showing a highly significant negative correlation with *Tyzzerella* (Fig. 5).

## Machine learning analysis based on NC_Group and CD_Group

Given that the taxonomic resolution of 16S rRNA sequencing is limited to the genus level, this study utilized genus-level relative abundance data of bacterial communities to screen for biomarkers (characteristic genera) by applying six classification models: SVM, XGBoost, RF, SLR, NN, and LR. The performance evaluation of these classification models (Fig. 6a; Table S3) indicated that, based on genus-level feature variables and confusion matrix analysis, the XGBoost model performed best, achieving an area under the curve (AUC) of 0.91. The remaining models ranked in performance as follows: RF, SVM, SLR, NN, and LR. In the XGBoost model, the top-ranked genera in terms of feature importance included *Tyzzerella*, *Lachnoclostridium*, *Paludicola*, *Escherichia–Shigella*, *Clostridium sensu stricto*, and *Veillonella*. In contrast, both the RF and SVM models identified *Tyzzerella* as the genus with the highest feature importance. These findings are consistent with the results of the fecal microbiota diversity analysis, suggesting that *Tyzzerella* may represent a core characteristic genus mediating the occurrence of diarrhea (Fig. 6b).

## Temporal dynamics of the gut microbiota at 10, 15, and 20 days of diarrhea

Analysis of microbial community dynamics revealed that at 10 days of diarrhea, α-diversity indices showed significant increases: the Chao1 and observed species indices at days 5 and 10 were significantly higher than those at day 0, while the Shannon and Simpson indices were markedly higher at day 10 compared with day 0 (Fig. 7a). Regarding β-diversity, principal coordinate analysis (PCoA) based on weighted UniFrac distances demonstrated clear clustering among groups (Fig. 7b), whereas no significant clustering was observed with unweighted UniFrac distances (Fig. 7c). At the phylum level, the dominant taxa included Firmicutes, Proteobacteria, and Fusobacteriota (Fig. 7d). Notably, Firmicutes were significantly more abundant at day 10 compared with days 0 and 5. Fusobacteriota were significantly more abundant at day 10 than at day 0, while Proteobacteria were significantly more abundant at day 0 and day 5 compared with day 10 (Fig. 7f). At the genus level, dominant taxa included *Escherichia–Shigella*, *Fusobacterium*, *Lactobacillus*, and Bacteroides (Fig. 7e). *Escherichia–Shigella* was significantly more abundant at day 0 compared with days 5 and 10, whereas *Faecalibacterium* was significantly more abundant at days 5 and 10 compared with day 0 (Fig. 7g). LefSe analysis (LDA threshold = 4) further identified *Escherichia–Shigella* and *Streptococcus* as characteristic genera in the CD_10d_0d comparison, *Bacteroides* and *Faecalibacterium* in the CD_10d_5d comparison, and *Lactobacillus*, *Fusobacterium*, and *Tyzzerella* in the CD_10d_10d comparison (Fig. 7h).

Microbial community dynamics were evaluated in calves with diarrhea at 15 days. For α-diversity, the Chao1, observed species, Shannon, and Simpson indices at day 0 were markedly higher than those at subsequent time points (Fig. 8a). β-diversity analysis based on both weighted UniFrac and unweighted UniFrac distances revealed distinct clustering of samples from day 0 compared with other groups (Fig. 8b). At the phylum level, Firmicutes, Proteobacteria, Fusobacteriota, and Bacteroidota were dominant (Fig. 8c). Firmicutes were significantly enriched at day 15 relative to day 0, whereas Proteobacteria were most abundant at day 0 and decreased progressively thereafter (Fig. 8e). At the genus level, *Escherichia–Shigella*, *Lactobacillus*, *Fusobacterium*, and *Streptococcus* predominated (Fig. 8d). *Escherichia–Shigella* abundance was highest at day 0 and declined over time, whereas *Lactobacillus* increased steadily with calf growth (Fig. 8f). LefSe analysis (LDA score = 4) identified *Escherichia–Shigella*, *Streptococcus*, and *Clostridium sensu stricto 1* as discriminative taxa in the CD_15d_0d group; *Peptostreptococcus*, *Lachnoclostridium*, *Bifidobacterium*, *Ruminococcus gnavus group*, *Butyricicoccus*, and *Bacteroides* in the CD_15d_5d group; *Lactobacillus*, *Faecalibacterium*, *Subdoligranulum*, *Akkermansia*, and *Ruminococcus torques group* in the CD_15d_10d group; and *Tyzzerella* as the sole discriminative genus in the CD_15d_15d group (Fig. 8g).

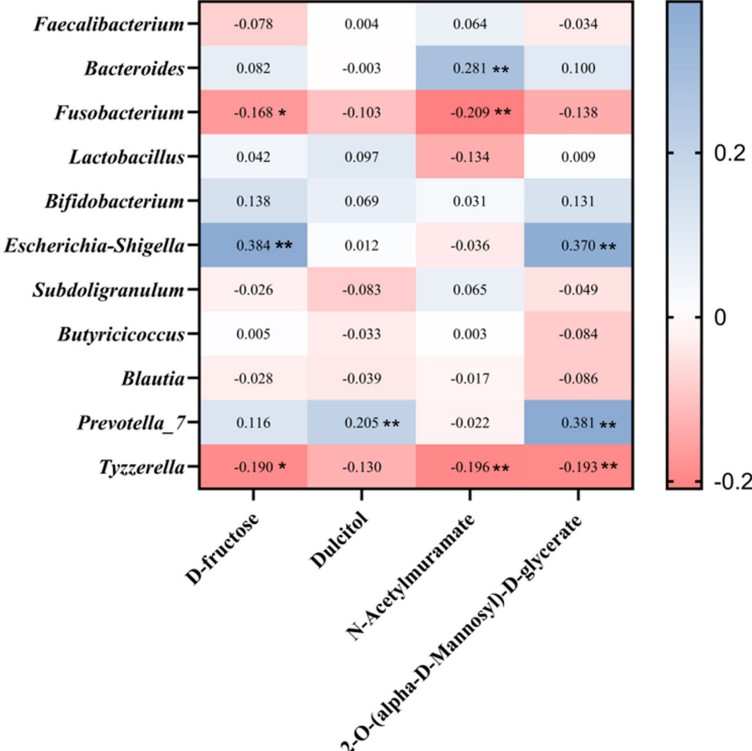

**FIG 5** Pearson correlation between microbial and metabolite levels under normal and diarrheal conditions in negative ion mode. The asterisks denote significant *P* values: *\*P < 0.05; **P < 0.01; ***P < 0.001.

Microbial community dynamics were further assessed at 20 days when diarrhea occurred. For α-diversity, the Chao1, observed species, Shannon, and Simpson indices at day 0 were markedly lower than those at subsequent time points, and all indices increased with sampling time (Fig. 9a). β-Diversity analysis revealed distinct clustering of samples from day 0 compared with the other groups (Fig. 9b). Taxonomic profiling at both phylum and genus levels indicated a significant increase in community richness with calf age (Fig. 9c and d). At the phylum level, Firmicutes were significantly less abundant at 0 day than at later time points, whereas Proteobacteria were significantly enriched at 0 day (Fig. 9e). At the genus level, *Escherichia–Shigella* exhibited the highest relative abundance at 0 day compared with other groups (Fig. 9f). LefSe analysis (LDA score = 4) identified *Escherichia–Shigella*, *Streptococcus*, and *Clostridium sensu stricto 1* as discriminative genera in the CD_20d_0d group; *Bacteroides*, *Butyricicoccus*, *Ruminococcus gnavus group*, *Peptostreptococcus*, *Lachnoclostridium*, and *Ligilactobacillus* in the CD_20d_5d group; *Lactobacillus* and *Akkermansia* in the CD_20d_10d group; *Faecalibacterium* and *Parabacteroides* in the CD_20d_15d group; and *Bifidobacterium*, *Blautia*, and *Tyzzerella* in the CD_20d_20d group (Fig. 9g).

## Metabolomics dynamic changes based on diarrhea days of 10, 15, and 20

At 10 days when diarrhea occurred, the numbers of up- and downregulated metabolites under positive and negative ionization modes are presented in Table S4. Enrichment analysis of differential metabolites indicated that, in the negative ion mode, the major enriched pathway was C5-branched dibasic acid metabolism, in which L-glutamic acid exhibited a continuous increasing trend, whereas citraconic acid decreased consistently (Fig. S2a). At 15 days, the numbers of up and downregulated metabolites under both ionization modes are summarized in Table S5. Pathway enrichment analysis revealed that, in the positive ion mode, differential metabolites were mainly enriched

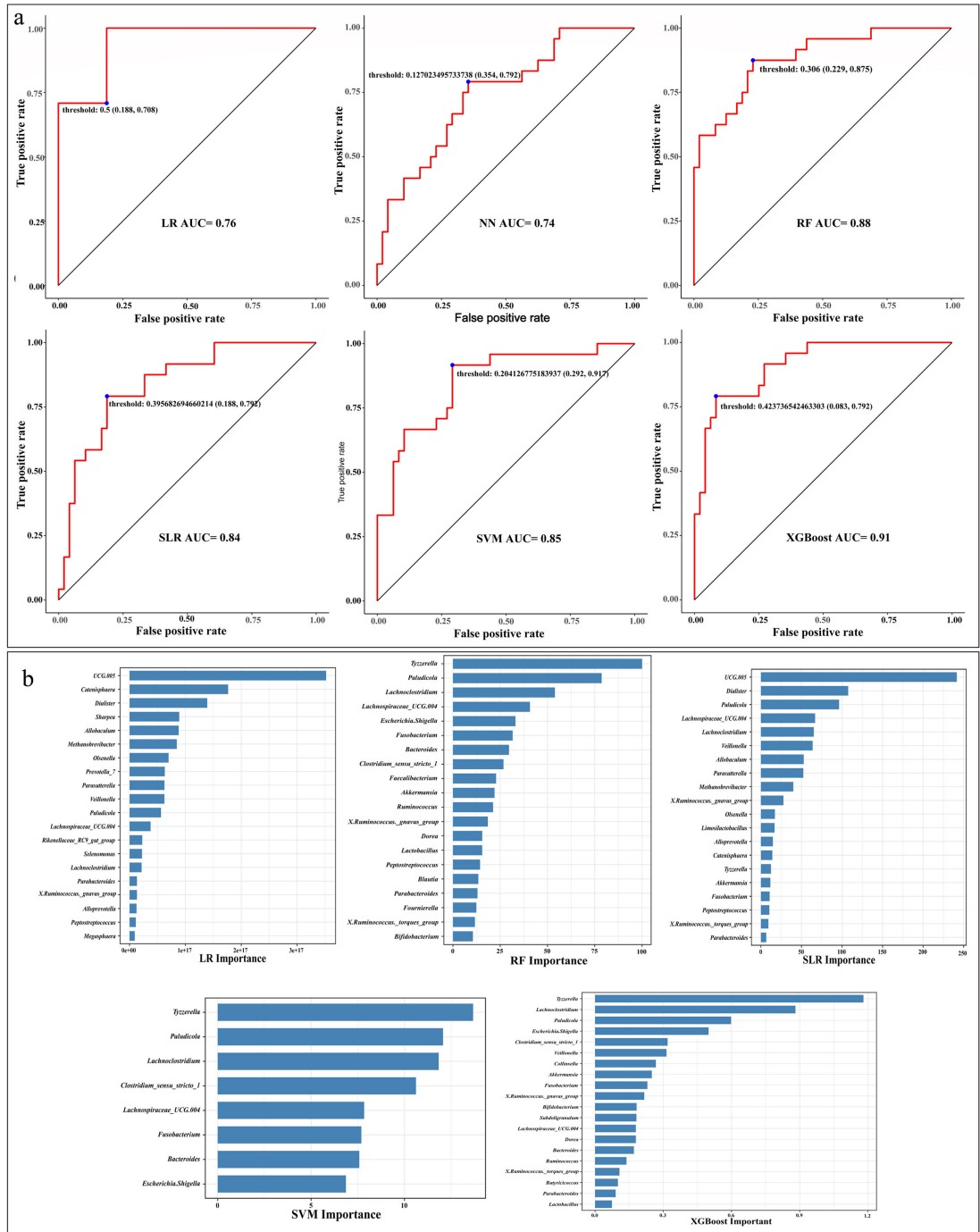

**FIG 6** Normal and diarrhea machine learning analysis. (a) ROC curves of six classification models. (b) Analysis of feature importance for five classification models.

in glycerophospholipid metabolism, tryptophan metabolism, and arginine and proline metabolism. Among these, (indol-3-yl) acetamide, 4-acetamidobutanoate, and indole-3-pyruvic acid initially increased but declined by 10 days (Fig. S2b), while 4-guanidino-butanoate, N-acetylputrescine, and picolinic acid peaked at 10 days and decreased thereafter at 15 days (Fig. S2c). Conversely, choline, L-arginine, and several lysophospha-tidylcholines (LysoPCs) decreased to their lowest levels at 10 days but increased again at 15 days (Fig. S2d). Indole, indolelactic acid, and tryptophan reached their highest abundances at 5 days, declined at 10 days, and increased again at 15 days (Fig. S2e). In the negative ion mode, the enriched pathways included degradation of aromatic

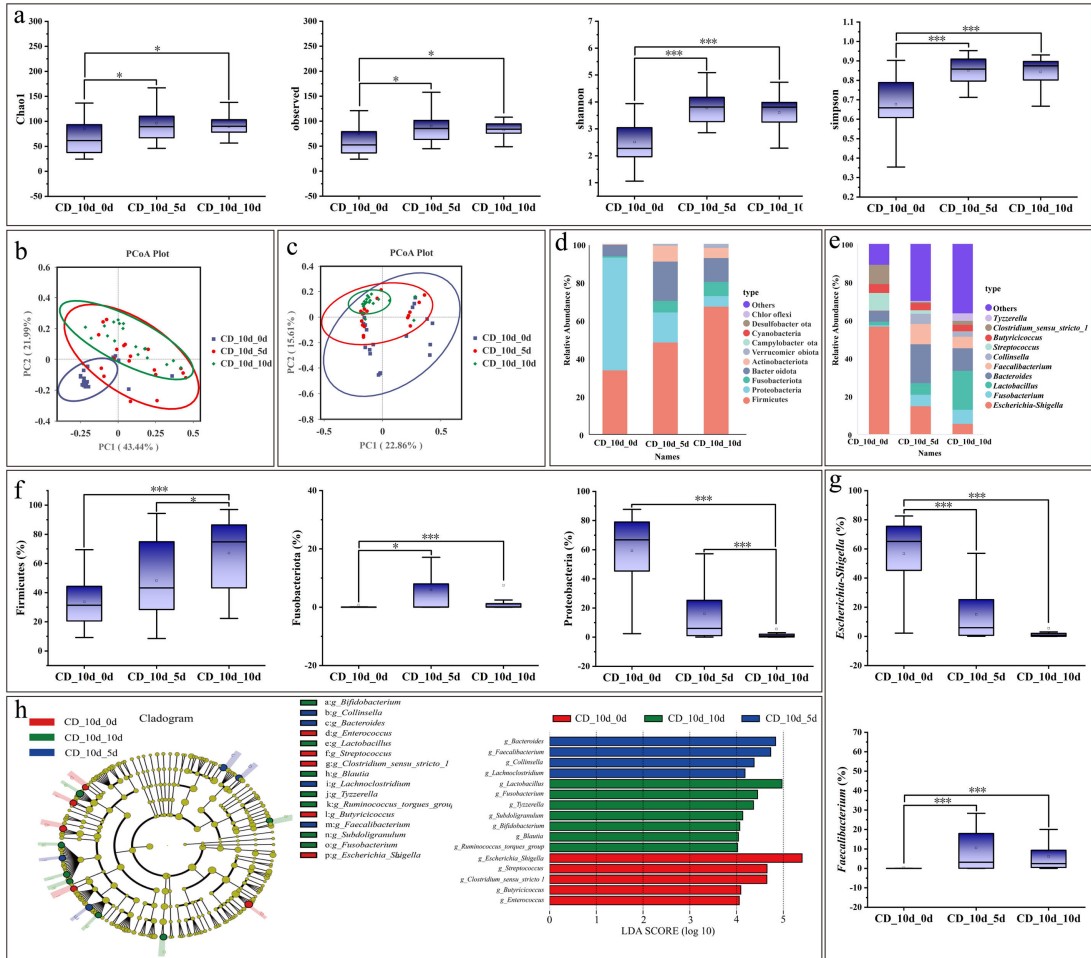

**FIG 7** Microbiological analysis during diarrhea on 10 days. (a) α-Diversity analysis. (b) β-Diversity—weighted_Unifrac PCoA plot. (c) β-Diversity—unweighted_Unifrac PCoA plot. (d) Relative abundance of bacterial phyla. (e) Relative abundance of bacterial genera. (f) Phylum-level taxon diversity. (g) Genus-level taxon diversity. (h) LefSe analysis. The evolutionary branching diagram is shown on the left. The LDA score graph is shown on the right. The asterisks denote significant *P* values: *P < 0.05; **P < 0.01; ***P < 0.001. Box plots show the median, upper and lower quartiles, and whiskers correspond to 1.5 times the interquartile range.

compounds, 2-oxocarboxylic acid metabolism, tryptophan metabolism, and pentose and glucuronate interconversions. Specifically, 10 metabolites, such as 2-formamino-benzoylacetate, 2-hydroxy-3-carboxy-6-oxo-7-methylocta-2,4-dienoate, 2-oxosuberate, 3-(6′-methylthio) hexylmalic acid, and phthalic acid, peaked at 10 days and declined at 15 days (Fig. S3a). Five metabolites, including 2-O-(α-D-mannosyl)-D-glycerate, 5-hydroxy-tryptophan, L-isoleucine, and L-rhamnonate, increased initially, peaked at 10 days, and decreased thereafter (Fig. S3b). Glucuronic acid and D-xylonic acid showed continuous decreases (Fig. S3c), whereas m-coumaric acid increased progressively (Fig. S3d). At 20 days, the numbers of up- and downregulated metabolites under both ionization modes are shown in Table S6. Enrichment analysis indicated that, in the positive ion mode, differential metabolites were mainly enriched in glycerophospholipid metabolism. Several LysoPCs and choline decreased to their lowest levels at 15 days and increased again at 20 days (Fig. S4a). Indole, indolelactic acid, and tryptophol peaked at 10 days, decreased thereafter, and showed a gradual increase again (Fig. S4b). In the negative ion mode, the enriched pathways included PTS, pentose and glucuronate interconversions, and fructose and mannose metabolism. Eight metabolites, such as 2-(acetamidomethylene) succinate, 2-formaminobenzoylacetate, 2-hydroxy-3-carboxy-6-oxo-7-methylocta-2,4-dienoate, and 2-oxosuberate, peaked at 15 days and decreased at

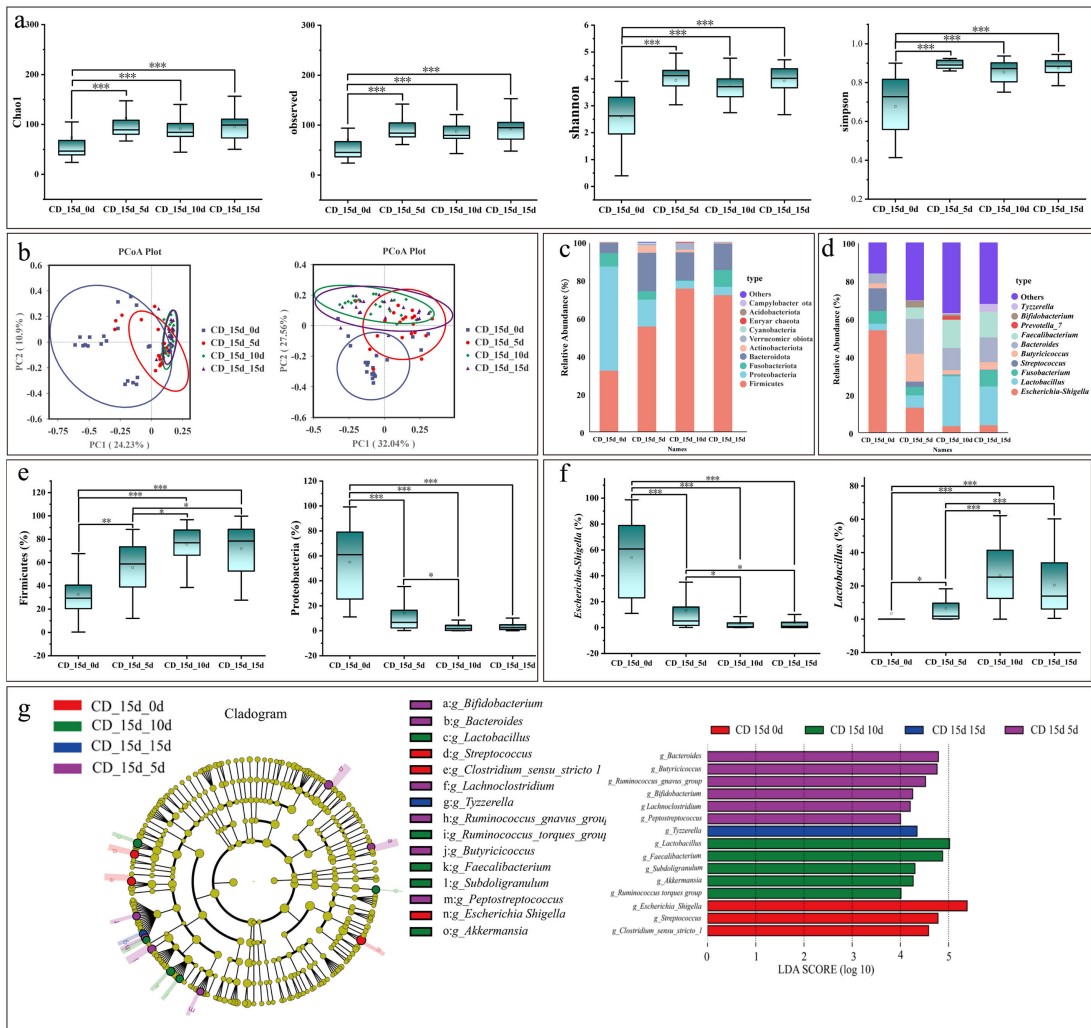

**FIG 8** Microbiological analysis during diarrhea on 15 days. (a) α-Diversity analysis. (b) β-Diversity—weighted_Unifrac PCoA plot (left) and unweighted _Unifrac PCoA plot (right). (c) Relative abundance of bacterial phyla. (d) Relative abundance of bacterial genera. (e) Phylum-level taxon diversity. (f) Genus-level taxon diversity. (g) LefSe analysis. The evolutionary branching diagram is shown on the left. The LDA score graph is shown on the right. The asterisks denote significant *P* values: *$P < 0.05$; **$P < 0.01$; ***$P < 0.001$. Box plots show the median, upper and lower quartiles, and whiskers correspond to 1.5 times the interquartile range.

20 days (Fig. S5a). Four metabolites, including 5,6-DHET, citraconic acid, N-acetyl-L-glutamate 5-semialdehyde, and prostaglandin E2, increased initially, peaked at 5 days, and decreased thereafter (Fig. S4b). Five metabolites, including phenethylamine glucuronide, N-carbamoylsarcosine, L-rhamnonate, and D-fructose, peaked at 10 days and decreased subsequently (Fig. S4c). D-xylonic acid and D-glucuronic acid exhibited continuous decreases (Fig. S4d), while m-coumaric acid displayed a continuous increasing trend (Fig. S4e). Moreover, under both positive and negative ion modes, the number of upregulated metabolites in diarrheic calves consistently exceeded that of downregulated metabolites.

## DISCUSSION

Our longitudinal monitoring of calves every 5 days after birth, with immediate fecal sampling and clinical recording upon diarrhea onset, demonstrated that diarrheic calves at 10, 15, and 20 days consistently exhibited significantly reduced fecal microbiota α-diversity (Chao1, Shannon, and observed species indices) compared with their healthy states prior to diarrhea. The reduction in microbial diversity observed here suggests that impaired gut microbial stability is closely associated with the onset of

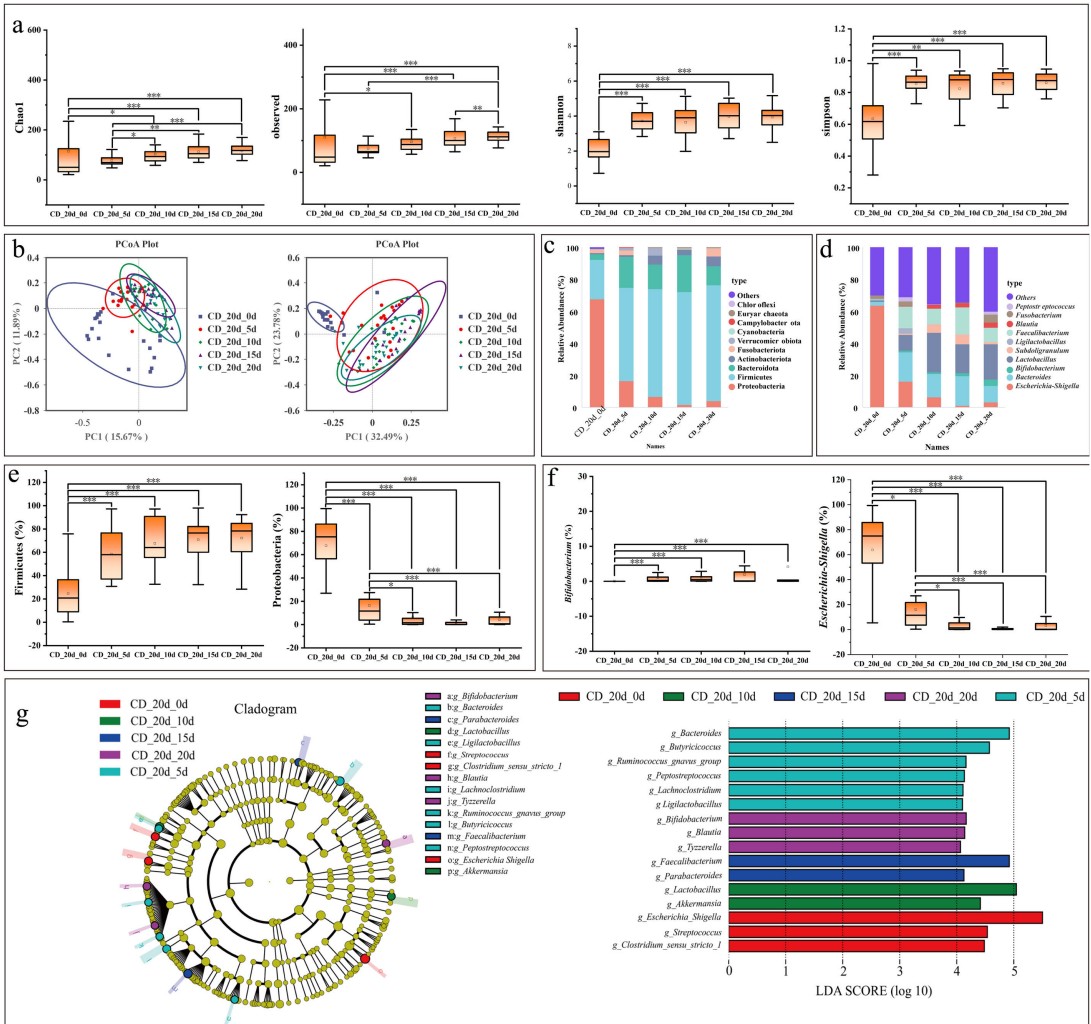

**FIG 9** Microbiological analysis during diarrhea on 20 days. (a) α-Diversity analysis. (b) β-Diversity—weighted_Unifrac PCoA plot (left) and unweighted _Unifrac PCoA plot (right). (c) Relative abundance of bacterial phyla. (d) Relative abundance of bacterial genera. (e) Phylum-level taxon diversity. (f) Genus-level taxon diversity. (g) LefSe analysis. The evolutionary branching diagram is shown on the left. The LDA score graph is shown on the right. The asterisks denote significant *P* values: \**P* < 0.05; \*\**P* < 0.01; \*\*\**P* < 0.001. Box plots show the median, upper and lower quartiles, and whiskers correspond to 1.5 times the interquartile range.

diarrhea. This pattern is in agreement with the findings of Zhu et al. in diarrheic piglets (26) and further supports the conclusions of Shi et al. in calves (27), indicating that reduced microbial diversity may represent a common microbial signature underlying diarrhea across different livestock species. At the same time, the reduction in overall diversity coexists with the emergence of specific diarrhea-enriched microbes. While β-diversity metrics primarily quantify global differences in community composition between samples, they may not fully capture abundance shifts of specific taxa that do not result in large-scale community restructuring. β-Diversity methods compare overall community profiles and thus can overlook subtle but biologically meaningful changes in individual taxa (28, 29). At the phylum level, the Firmicutes-to-Bacteroidota ratio in the NC group was significantly lower than that in the CD group, which was consistent with the trend reported by Zhu et al. (26). Multiple studies have suggested that an increased Firmicutes/Bacteroidota ratio is positively associated with the risk of diarrhea and disease occurrence in animals (30–32), and a similar pattern was observed in the present study. Moreover, the relative abundance of Bacteroidota was significantly higher in the NC group than in the CD group, in agreement with the findings of Cao et al. (33). The reduction in Bacteroidota may be related to intestinal microbial dysbiosis,

environmental changes, and pathological mechanisms. Verrucomicrobiota, although a low-abundance phylum in the gut, plays a critical functional role, with its representative species *Akkermansia muciniphila* being essential for maintaining the intestinal barrier and regulating host metabolism. In the present study, the abundance of Verrucomicrobiota was significantly higher in the NC group. By contrast, Fusobacteriota, a group of Gram-negative, anaerobic, spindle-shaped bacteria, was markedly more abundant in the CD group than in the NC group. These findings suggest that *A. muciniphila* may exert a protective role against diarrhea by supporting mucosal integrity and metabolic homeostasis, whereas the enrichment of Fusobacteriota in diarrheic calves may be linked to inflammatory processes or pathogenic mechanisms associated with gut dysbiosis. At the genus level, *Fusobacterium* exhibited a similar trend, consistent with previous studies (34–36). Its pathogenic mechanisms may involve the modulation of tight junction proteins (e.g., Zonula Occludens-1 and Occludin), leading to disruption of epithelial integrity and increased intestinal permeability. In addition, *Fusobacterium* can promote cytokine secretion, activate the STAT3 signaling pathway, and induce CD4[+] T cell proliferation and Th1/Th17 immune responses, thereby contributing to barrier dysfunction and aberrant inflammation (37). Other studies have reported that *Fusobacterium* activates Toll-like receptor (TLR) signaling, stimulating colonic epithelial cells to secrete proinflammatory factors, which results in mucus layer depletion and immune cell infiltration (38). In cases of severe infection, it may even penetrate the gut-blood barrier, causing systemic manifestations such as sepsis and severe watery diarrhea (39). In contrast, several beneficial genera, including *Subdoligranulum* (40), *Faecalibacterium* (41), and *Bacteroides* (42), were significantly more abundant in the NC group. The lack of clear separation in β-diversity space suggests that diarrhea in these calves does not result in a complete restructuring of the gut ecosystem, but rather a "dysregulation" of specific keystone taxa within a largely stable microbial framework. This highlights *Fusobacterium* and *Tyzzerella* not as drivers of wholesale community change, but as sensitive indicators of localized ecological disturbance.

LEfSe analysis indicated that *Fusobacterium* and *Tyzzerella* were significantly enriched in diarrheic calves. *Tyzzerella* is a genus of Gram-negative anaerobic bacilli, among which *T. piliformis* is the most representative pathogenic species and has been closely associated with gastrointestinal diarrheal symptoms in various animals and occasionally in humans (43). Infection with this species can cause Tyzzer's disease, which typically manifests as watery or mucoid diarrhea, particularly in young or immunocompromised hosts (44). Evidence from animal models, such as guinea pigs (45) and rabbits (46), has demonstrated that infection with *T. piliformis* leads to necrotizing enteritis, severe diarrhea, and even mortality, confirming its pathogenic potential. In the present study, the abundance of *Tyzzerella* was markedly elevated in diarrheic calves, suggesting a potential association with the onset of diarrhea. *Fusobacterium*, another Gram-negative anaerobic genus, may contribute to diarrhea through the induction of intestinal inflammatory responses. Previous studies have shown that *Fusobacterium* can activate the NLRP3 inflammasome in macrophages and epithelial cells, promoting the release of inflammatory cytokines, such as IL-1β, and exacerbating intestinal injury (47). The enrichment of *Fusobacterium* observed in diarrheic calves in this study further implies that it may disrupt epithelial tight junctions, induce apoptosis, and compromise mucosal integrity via these inflammation-mediated mechanisms. Nonetheless, the precise pathogenic pathways require further investigation.

Preliminary microbiome analyses comparing diarrheic and healthy calves suggested that microbial metabolic pathways may play a key role in the pathogenesis of calf diarrhea. To further investigate this, we performed fecal metabolomic profiling to assess metabolic dysfunction in diarrheic calves. The results revealed significant differences in the PTS metabolic pathway between the two groups under the negative ion mode, with core metabolites enriched in this pathway including dulcitol, 2-O-(α-D-mannosyl)-D-glycerate, N-acetylmuramate, and D-fructose. Among these, dulcitol, a polyol with low metabolic efficiency, can escape complete absorption in the small intestine,

thereby increasing osmotic pressure in the intestinal lumen. This osmotic imbalance promotes water influx into the gut, leading to luminal distension, accelerated peristalsis, and ultimately diarrhea (48). 2-O-(α-D-mannosyl)-D-glycerate, a naturally occurring low-molecular-weight glycosylated metabolite belonging to the family of compatible solutes, may contribute to osmotic diarrhea by accumulating in the intestinal lumen and increasing osmotic pressure (49). N-acetylmuramate (MurNAc) and its precursor can activate the NOD2 receptor, triggering the NF-κB pathway and promoting the release of proinflammatory cytokines, such as TNF-α and IL-1β, thereby impairing intestinal barrier function and leading to inflammatory diarrhea. Notably, NOD2-deficient mice develop dysbiosis and inflammatory diarrhea due to impaired microbial regulation (50, 51), suggesting a close association between MurNAc abundance and diarrheal outcomes. D-fructose, a common monosaccharide, can also induce diarrhea when intake exceeds the absorptive capacity of the small intestinal GLUT5 transporter. The unabsorbed fraction undergoes microbial fermentation in the colon, generating short-chain fatty acids (SCFAs), hydrogen, and other metabolites that provoke diarrheal symptoms. Fructose malabsorption is also recognized as a frequent cause of diarrhea and abdominal pain in children (52–54). Taken together, dulcitol, 2-O-(α-D-mannosyl)-D-glycerate, N-acetylmuramate, and D-fructose may represent key metabolites mediating calf diarrhea, and the PTS metabolic pathway may be implicated in its development, although the precise regulatory mechanisms require further investigation.

Pearson correlation analysis revealed that the relative abundances of *Tyzzerella* and *Fusobacterium* were significantly negatively correlated with several key metabolites, including dulcitol, 2-O-(α-D-mannosyl)-D-glycerate, N-acetylmuramate, and D-fructose. Although correlation does not imply causation, these findings suggest that these two genera may contribute to the development of calf diarrhea by reshaping the luminal metabolic milieu and impairing intestinal mucosal barrier function. Previous studies have consistently reported that *Fusobacterium* is closely associated with enhanced mucosal inflammation, increased epithelial permeability, and disruption of tight junction integrity, thereby promoting inflammatory responses and aggravating barrier damage and ultimately forming a vicious cycle of "barrier impairment–inflammation exacerbation-symptom aggravation" (55–57). In contrast, *Tyzzerella* may serve as an inflammation-associated ecological indicator and metabolic network node, and its increased abundance may reflect a competitive advantage under diarrhea-induced alterations of the intestinal environment (58). The reduced levels of carbohydrate-related metabolites, such as D-fructose and dulcitol, may be attributable to accelerated intestinal transit, altered sites of nutrient absorption and fermentation, and competitive microbial utilization of carbon sources during diarrhea (59). Meanwhile, barrier dysfunction and inflammation may further contribute to luminal metabolite dilution or depletion. N-Acetylmuramate, a core component of bacterial peptidoglycan, may reflect changes in bacterial cell wall turnover and host innate immune signaling (60). In addition, 2-O-(α-D-mannosyl)-D-glycerate, a carbohydrate derivative, may indicate alterations in microbial carbohydrate metabolism and osmoadaptation pathways under diarrheal conditions.

In the present study, fecal microbiota data at the genus level were used to construct and evaluate SVM, XGBoost, RF, SLR, NN, and LR classification models for biomarker screening. Among these, *Tyzzerella* was identified as the genus with the highest importance in the XGBoost, RF, and SVM models, with XGBoost showing the best overall performance. Consistent with these computational findings, microbiological analyses also demonstrated that *Tyzzerella* was a major contributor to diarrhea, in line with previous reports that have implicated this genus as a causative factor in diarrheal disease (43–46). Moreover, *Fusobacterium* consistently ranked highly across multiple models, further highlighting its potential role as a key bacterial genus associated with diarrhea.

The present study found that at 10 days, 15 days, and 20 days, diarrheic calves exhibited significantly higher α-diversity indices (Chao1, Shannon, observed species, and Simpson) compared with those at 0 day, consistent with the findings of Ma and Chen et al. (61, 62). For β-diversity, both weighted UniFrac and unweighted

UniFrac analyses showed clear clustering of samples at 0 day relative to other time points. At the phylum level, *Firmicutes* increased progressively with calf age, whereas *Proteobacteria* displayed the opposite trend, in agreement with the results of Chen et al. (62). At the genus level, *Escherichia–Shigella* showed higher abundance at 0 day than at subsequent time points. This pattern may be explained by several factors: (i) the "aerobic–anaerobic succession" theory of early colonization, whereby facultative anaerobes (e.g., *Escherichia coli*) initially colonize the oxygen-rich neonatal gut, consume oxygen, and create an anaerobic environment favorable for strict anaerobes (63); (ii) the oxygen gradient–driven succession mechanism, in which the neonatal gut transitions from aerobic to microaerobic and ultimately anaerobic conditions, facilitating the establishment of facultative anaerobes as pioneers (64), with facultative anaerobes consuming oxygen through respiration to promote colonization by strict anaerobes (65); and (iii) the association of *Escherichia–Shigella* with diarrhea, where pathogenic *E. coli* abundance may decline during diarrheal episodes due to host immune clearance or competitive inhibition (66). The significantly lower microbial diversity observed at 0 day at both phylum and genus levels is likely attributable to neonatal physiology, as the calf gut is nearly sterile at birth and microbial colonization requires time to establish. Early low diversity reflects an incompletely developed microbial community, which gradually increases with age and environmental exposure (67). Furthermore, reliance on milk—especially colostrum—as the sole nutrient source during the early stage may restrict microbial diversity due to the antimicrobial properties of milk components such as lactoferrin and immunoglobulins. The introduction of solid feed markedly enhances microbial diversity (68). Additionally, LefSe analysis across different diarrheal time points, combined with results from both machine learning and microbiological analyses, consistently highlighted *Tyzzerella* as a genus of particular concern in relation to calf diarrhea.

Dynamic changes in metabolites are closely associated with disease onset, progression, and resolution, as imbalances in metabolic networks reflect abnormalities in cellular function, signaling pathways, and the microenvironment. Such dynamic biomarkers provide valuable opportunities for early disease warning, therapeutic monitoring, and personalized interventions (69). In this study, we found that in calves experiencing diarrhea at 10 days, L-glutamic acid exhibited an upward trend under the negative ion mode and peaked at the time of diarrhea, whereas citraconic acid followed the opposite pattern, reaching its lowest level during diarrheal episodes. Both metabolites were enriched in the C5-branched dibasic acid metabolism pathway, which is involved in intestinal absorption, metabolism, and immune regulation. Their pronounced alterations during diarrhea highlight their potential value as candidate biomarkers for calf diarrhea.

At 15 and 20 days, during diarrheal episodes, choline and two lysophosphatidylcholines (LysoPC(22:5(4Z,7Z,10Z,13Z,16Z)/0:0) and LysoPC(15:0/0:0)) under the positive ion mode exhibited similar dynamics, decreasing to their lowest levels prior to diarrhea and rising significantly thereafter. Tryptophol peaked at day 5, declined subsequently, and increased again at the onset of diarrhea. In the negative ion mode, five metabolites—2-formaminobenzoylacetate, 2-hydroxy-3-carboxy-6-oxo-7-methylocta-2,4-dienoate, 2-oxosuberate, 3-(6′-methylthio)hexylmalic acid, and phthalic acid—rose to peak levels prior to diarrhea but dropped markedly at its onset. By contrast, glucuronic acid showed a continuous decline, whereas m-coumaric acid exhibited the opposite trend. Mechanistically, diarrheal episodes accompanied by microbial dysbiosis (e.g., loss of beneficial taxa) may disrupt choline metabolism, reducing its absorption and promoting direct excretion (70). As choline is converted into acetylcholine to regulate intestinal peristalsis, accelerated peristalsis during diarrhea may further shorten absorption time and enhance excretion (71). LysoPCs, as critical intermediates of phospholipid metabolism, can trigger diarrhea at pathological concentrations by compromising mucosal integrity. Tryptophol, a microbial metabolite of tryptophan, can damage the mucus layer and epithelial barrier, leading to reduced MUC2 protein levels, decreased mucus thickness, and subsequent osmotic diarrhea (72). For

other metabolites, 2-formaminobenzoylacetate, a product of phenylalanine metabolism, declined at diarrhea onset, possibly reflecting impaired microbial degradation of aromatic amino acids (73, 74). 2-Hydroxy-3-carboxy-6-oxo-7-methylocta-2,4-dienoate, a fatty acid oxidation product, decreased under conditions of oxidative stress and mitochondrial dysfunction during diarrhea (75). 2-Oxosuberate, involved in archaeal coenzyme biosynthesis, was reduced in association with decreased methanogen abundance under dysbiotic conditions (76). 3-(6′-Methylthio) hexylmalic acid, closely linked to sulfur metabolism, may indicate accumulation of pro-inflammatory sulfur-containing metabolites, such as hydrogen sulfide, and activation of the NF-κB pathway (76). Phthalic acid, an ester-related metabolite, increased prior to diarrhea, possibly reflecting barrier disruption and microbial metabolic imbalance (74). By contrast, glucuronic acid and m-coumaric acid exhibited continuous monotonic changes and are therefore unsuitable as predictive markers for diarrhea.

In summary, L-glutamic acid and citraconic acid during diarrhea at 10 days, as well as choline, LysoPC, 2-formaminobenzoylacetate, 2-hydroxy-3-carboxy-6-oxo-7-methylocta-2,4-dienoate, 2-oxosuberate, 3-(6′-methylthio) hexylmalic acid, and phthalic acid during diarrhea at 15 and 20 days, exhibited pronounced dynamic response patterns. These metabolites may serve as potential biomarkers for calf diarrhea, with promising implications for early warning and mechanistic elucidation.

## Conclusion

In this study, we systematically elucidated the dynamic alterations of gut microbiota and metabolites during calf diarrhea using multi-omics and machine learning approaches. The results demonstrated that diarrheic calves exhibited significantly reduced microbial diversity and pronounced structural dysbiosis, characterized by decreased abundances of *Bacteroidota* and *Verrucomicrobiota*, alongside enrichment of *Firmicutes*, *Fusobacteriota*, and diarrhea-associated pathogenic genera, such as *Tyzzerella* and *Fusobacterium*. Among these, *Tyzzerella* was validated by machine learning as the most predictive biomarker (AUC = 0.91). Metabolomic analysis revealed that differential metabolites were mainly enriched in pathways such as PTS, with key metabolites including dulcitol and N-acetylmuramate potentially contributing to diarrheal progression through osmotic or inflammatory mechanisms. Pearson correlation analysis further supported the association between *Tyzzerella* and differential metabolites, reinforcing its potential pathogenic role. Dynamic profiling also indicated that early high abundance of *Escherichia–Shigella* and the continuous shifts of *Firmicutes* and *Proteobacteria* were closely related to diarrhea onset, while metabolites such as L-glutamic acid, choline, and LysoPCs exhibited distinct accumulation patterns.

Despite limitations such as single-source sampling and time intervals, our study provides key microbial and metabolic targets for the early diagnosis and mechanistic understanding of calf diarrhea. Future research will adopt a more targeted approach, including higher-frequency longitudinal sampling (e.g., daily sampling or the use of automated barn-based health monitoring systems) to capture the exact timing of physiological transitions. This will help verify whether the biomarkers identified in our current model are sufficiently sensitive to predict rapidly developing cases. Additionally, future studies should adopt non-interventional observational designs (where ethically permissible) or controlled animal models to specifically investigate the microbial and metabolic characteristics associated with recovery phases and chronic dysbiosis. Large-scale, multicenter studies will be necessary to validate the generalizability of these biomarkers and further explore their underlying mechanisms.

## ACKNOWLEDGMENTS

This work was supported by the National Natural Science Foundation of China (32260898), China Agriculture Research System of MOF and MARA (CARS-37), Key Research and Development Project of Gansu Provincial Department of Science and

Technology (24YFNA016), Modern Cold and Drought Characteristic Agricultural Science and Technology Support Project of Gansu Province (KJZC-2025-14), and Gansu Province Basic Research Program-Special Excellent Doctoral Student Project of Natural Science Foundation (26JRRA653).

Xitong Yin, Conceptualization, Data curation, Writing-original draft, Writing-review and editing; Yanlong Niu, Data curation, Formal analysis, Methodology; Baoxia Chen, Conceptualization, Data curation; Hao Zhang, Data curation, Methodology; Rongxia Guo, Data curation; Chun Niu, Methodology; Jianguo Kang, Methodology; Hongmei Shi, Methodology; Xiangying Kong, Methodology; Weidong Ma, Conceptualization; Zhongfa Ma, Methodology; Yanming Wei, Methodology; Yongli Hua, Conceptualization, Funding acquisition, Writing-review and editing.

## AUTHOR AFFILIATIONS

[1]College of Veterinary Medicine, Gansu Agricultural University, Lanzhou, China
[2]Gansu Gannan Animal Husbandry and Veterinary Workstation, Gannan, China
[3]Qinghai Haibei Animal Husbandry and Veterinary Science Research Institute, Haibei, China
[4]Shaanxi Agricultural and Animal Husbandry Breeding Farm, Baoji, China
[5]Gansu Xin Gaoyuan Agriculture and Animal Husbandry Development Co., Ltd., Lanzhou, China

## AUTHOR ORCIDs

Xitong Yin http://orcid.org/0009-0009-8092-7431
Yongli Hua http://orcid.org/0009-0004-2666-6772

## AUTHOR CONTRIBUTIONS

Xitong Yin, Conceptualization, Data curation, Formal analysis, Writing – original draft, Writing – review and editing | Yanlong Niu, Data curation, Formal analysis, Methodology | Baoxia Chen, Conceptualization, Data curation | Hao Zhang, Data curation, Methodology | Rongxia Guo, Data curation | Chun Niu, Methodology | Jianguo Kang, Methodology | Hongmei Shi, Methodology | Xiangying Kong, Methodology | Weidong Ma, Conceptualization | Zhongfa Ma, Methodology | Yanming Wei, Methodology | Yongli Hua, Conceptualization, Funding acquisition, Writing – review and editing

## DATA AVAILABILITY

The 16S rRNA gene sequencing raw data have been deposited in the NCBI database (PRJNA1330053). The metabolomics data sets generated and analyzed during the current study are available in the OMIX repository of the National Genomics Data Center (NGDC), under accession number OMIX012239.

## ETHICS APPROVAL

All studies involving animals were carried out in accordance with the regulations for the Administration of Affairs Concerning Experimental Animals (Ministry of Science and Technology, China, revised in June 2004), and all experimental procedures were reviewed and approved by the Animal Protection and Use Committee of Gansu Agricultural University (Approval No: GSAU-Eth-VMC-2024-043).

## ADDITIONAL FILES

The following material is available online.

### Supplemental Material

**Supplemental material (mSystems00005-26-s0001.docx).** Fig. S1 to S5; Tables S1 to S6.

## Open Peer Review

**PEER REVIEW HISTORY (review-history.pdf).** An accounting of the reviewer comments and feedback.

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
