## [Reviewer comments · mSystems]

Screening and dynamic change study of microbial and metabolite markers for calf diarrhea based on multi-omics and machine learning

Xitong Yin, Yanlong Niu, Baoxia Chen, Hao Zhang, Rongxia Guo, Chun Niu, Jianguo Kang, Hongmei Shi, Xiangying Kong, Weidong Ma, Zhongfa Ma, Yanming Wei, and Yong-li Hua

Corresponding Author(s): Yong-li Hua, Gansu Agricultural University College of Veterinary Medicine

Review Timeline:

Submission Date:	January 5, 2026
Editorial Decision:	February 4, 2026
Revision Received:	February 25, 2026
Accepted:	March 3, 2026

Editor: Yu-Liang Yang

Reviewer(s): Disclosure of reviewer identity is with reference to reviewer comments included in decision letter(s). The following individuals involved in review of your submission have agreed to reveal their identity: Yuting Zhai (Reviewer #2)

Transaction Report:

DOI: <https://doi.org/10.1128/msystems.00005-26>

Re: mSystems00005-26 (**Screening and dynamic change study of microbial and metabolite markers for calf diarrhea based on multi-omics and machine learning**)

Dear Dr. Yong-li Hua:

Thank you for the privilege of reviewing your work. Below you will find the instructions from the mSystems editorial office, and the reviewer comments.

Revision Guidelines

Sincerely,
Yu-Liang Yang
Editor
mSystems

Reviewer #1 (Comments for the Author):

The paper titled "Screening and dynamic change study of microbial and metabolite markers for calf diarrhea based on multi-omics and machine learning", this study investigated the dynamic microbial and metabolic signatures of neonatal calf diarrhea using multi-omics and machine learning. Diarrheic calves showed significantly reduced gut microbial diversity. At the genus level, Tyzzerella and Fusobacterium were identified as core biomarkers, with Tyzzerella achieving high predictive accuracy (AUC=0.91) in an XGBoost model. Key metabolites like Dulcitol and N-Acetylmuramate were linked to diarrheal progression via osmotic and inflammatory pathways. Additionally, a high abundance of Escherichia-Shigella at birth was identified as a potential

early trigger. These findings provide a framework for early diagnosis and targeted precision interventions. The manuscript is well-written. The reviewer has only some minor concerns as follows :

1. In materials and methods (page 3), authors implemented strict criteria that may have omitted relevant biological data. The exclusion criteria in this study, calves that developed diarrhea only outside of the pre-scheduled sampling time points (days 0, 5, 10, 15, and 20) were excluded from the study. By ignoring calves that fell ill between these 5-day intervals, the researchers may have missed unique microbial or metabolic "early-warning" signals that occur on a more rapid timeline, potentially limiting the model's sensitivity to fast-onset cases. Authors could do a brief explanation may help the understanding the exclusion criteria of specific diarrheal cases.
2. In this study, once a calf presented with a fecal score of 2-3, it was treated by a veterinarian and was not scored or sampled for the trial thereafter. Ethically necessary for animal welfare, this prevents the study from mapping the "recovery phase" or the long-term impact of diarrhea on the gut microbiome. Consequently, the data focuses only on the onset and does not provide insights into which microbial markers indicate a successful return to health versus chronic dysbiosis. Authors could describe more details in these sections or in discussion.
3. In page 5 and 6, the results text, the authors state that α -diversity indices (Chao1, Shannon) were significantly higher in the Normal group. However, having 3x more "unique species" in the Diarrhea group typically suggests higher richness, which contradicts the conclusion that diversity is reduced in sick calves. You could challenge the authors on whether the "unique species" in the Diarrhea group are actually high-quality biological signals or merely transient environmental contaminants/sequencing noise that wasn't properly filtered. Discrepancy in OTUs, the Diarrhea group (CD_Group) is reported to have 2,475 unique species, while the Normal group (NC_Group) has only 798. The authors present a major difference in the number of "unique species" between the two groups, which may be a result of unequal sample sizes or sampling bias rather than a biological reality of the disease. Authors could clarify this in results or discussion part.
4. While this manuscript emphasizes clear differences in microbial composition, the actual spatial separation of the groups in the initial analysis is weak. The results state that β -diversity analysis (using Weighted and Unweighted Unifrac) "did not show clear separation between the two groups" in the initial PCoA. If the overall microbial communities do not cluster separately, it suggests that the "core" microbiome of a healthy calf and a diarrheic calf are more similar than the authors imply. You can challenge the validity of using *Tyzzera* and *Fusobacterium* as "robust" biomarkers when the global community structure fails to distinguish between a sick and healthy animal at a statistical level. Authors may clarify this in results or discussion parts.

Reviewer #2 (Comments for the Author):

This manuscript investigates changes in gut microbiota and fecal metabolites associated with neonatal calf diarrhea using longitudinal sampling, 16S rRNA sequencing, metabolomics, and machine-learning approaches. The integration of multi-omics data and ML algorithms to identify predictive biomarkers for calf diarrhea is of potential interest to the field. Overall, the study provides valuable data, and the analytical workflow is generally well structured and logically presented. However, several aspects, including sampling description, grouping logic, figure resolution, terminology consistency, and data presentation require clarification.

1. The sampling workflow and grouping strategy remain confusing.
 - The manuscript states that 20 samples were selected from each time point, resulting in 80 samples in total, but the numbers and grouping described in Supplementary Tables S1 and S2 suggest more samples were included at certain time points.
 - The logic for selecting the final 20 samples per time point is not sufficiently explained.
 - It is unclear whether the ratio between diarrheic and non-diarrheic calves is balanced across time points.
 - The grouping strategy (NRM_, CD_, etc.) is difficult to follow without clear explanation in the main text.
 - The authors may consider providing a flow chart summarizing the total calf, excluding calf, number sampled per time point and analytical groups.
2. Overall, all figures are having low resolutions, causing reading difficulties.
3. The top and bottom whiskers in the boxplots are not explained.
4. If relative abundance ranges from 0-100, the authors should include percentage (%) in x-axis title.
5. Consider replacing vague legend terms like "type" with specific taxonomic level names.
6. Line 20: *Escherichia-Shigella* should be italic
7. Figure 1: Figure legend is missing.
8. Lines 105-106: How the 20 samples were selected and whether group ratios are balanced (as mentioned above).
9. Most places in the manuscript used "normal vs diarrhea"; in figure 2 it says "No diarrhea and diarrhea"
10. Supplementary Table 1: Provide explanation for abbreviations and numbering scheme.
11. Supplementary Table 2: Suggest replacing "will experience diarrhea" with "experienced diarrhea".
12. Clarify why CD_Group excludes certain early time points.

13. Sampling illustration and grouping tables are overly complex and need simplification.
14. Line 125: qPCR
15. Line 200: Chao 1
16. Line 216: "phylum level species" is confusing, consider using "phylum-level taxa".
17. Line 250: "a-d"
18. Line 250: consider using "The left panel", "the right panel"
19. Remove unintended highlighting in Supplementary Tables 4-6.
20. Line 465-471: How the linkage between microbiome and metabolome influence phenotypical differences of calves are not fully discussed, suggest to strength this part of discussion.

Comments and Suggestions

The paper titled "Screening and dynamic change study of microbial and metabolite markers for calf diarrhea based on multi-omics and machine learning", this study investigated the dynamic microbial and metabolic signatures of neonatal calf diarrhea using multi-omics and machine learning. Diarrheic calves showed significantly reduced gut microbial diversity. At the genus level, Tyzzerella and Fusobacterium were identified as core biomarkers, with Tyzzerella achieving high predictive accuracy (AUC=0.91) in an XGBoost model. Key metabolites like Dulcitol and N-Acetylmuramate were linked to diarrheal progression via osmotic and inflammatory pathways. Additionally, a high abundance of Escherichia-Shigella at birth was identified as a potential early trigger. These findings provide a framework for early diagnosis and targeted precision interventions. The manuscript is

well-written. The reviewer has only some minor concerns as follows :

1. In materials and methods (page 3), authors implemented strict criteria that may have omitted relevant biological data. The exclusion criteria in this study, calves that developed diarrhea only outside of the pre-scheduled sampling time points (days 0, 5, 10, 15, and 20) were excluded from the study. By ignoring calves that fell ill between these 5-day intervals, the researchers may have missed unique microbial or metabolic "early-warning" signals that occur on a more rapid timeline, potentially limiting the model's sensitivity to fast-onset cases. Authors could do a brief explanation may help the understanding the exclusion criteria of specific diarrheal cases.
2. In this study, once a calf presented with a fecal score of 2–3, it was treated by a veterinarian and was not scored or sampled for the trial thereafter. Ethically necessary for animal welfare, this prevents the study from mapping the "recovery phase" or the long-term impact of diarrhea on the gut microbiome. Consequently, the data focuses only on the onset and does not provide insights into which microbial markers indicate a successful return to health versus chronic dysbiosis. Authors could describe more details in these sections or in discussion.
3. In page 5 and 6, the results text, the authors state that alpha-diversity indices (Chao1, Shannon) were significantly higher in the Normal group. However, having 3x more "unique species" in the Diarrhea group typically suggests higher richness, which contradicts the conclusion that diversity is reduced in sick calves. You could challenge the authors on whether the "unique species" in the Diarrhea group are

actually high-quality biological signals or merely transient environmental contaminants/sequencing noise that wasn't properly filtered. Discrepancy in OTUs, the Diarrhea group (CD_Group) is reported to have 2,475 unique species, while the Normal group (NC_Group) has only 798. The authors present a major difference in the number of "unique species" between the two groups, which may be a result of unequal sample sizes or sampling bias rather than a biological reality of the disease. Authors could clarify this in results or discussion part.

4. While this manuscript emphasizes clear differences in microbial composition, the actual spatial separation of the groups in the initial analysis is weak. The results state that beta-diversity analysis (using Weighted and Unweighted Unifrac) "did not show clear separation between the two groups" in the initial PCoA. If the overall microbial communities do not cluster separately, it suggests that the "core" microbiome of a healthy calf and a diarrheic calf are more similar than the authors imply. You can challenge the validity of using *Tyzzarella* and *Fusobacterium* as "robust" biomarkers when the global community structure fails to distinguish between a sick and healthy animal at a statistical level. Authors may clarify this in results or discussion parts.

Dear Reviewers

We would like to thank you very much for your valuable comments and good suggestions that greatly helped to improve our manuscript. We have carefully considered your valuable comments and good suggestions. In the following, we are going to explain how your comments have been taken into full account in the revision.

1. In materials and methods (page 3), authors implemented strict criteria that may have omitted relevant biological data. The exclusion criteria in this study, calves that developed diarrhea only outside of the pre-scheduled sampling time points (days 0, 5, 10, 15, and 20) were excluded from the study. By ignoring calves that fell ill between these 5-day intervals, the researchers may have missed unique microbial or metabolic "early-warning" signals that occur on a more rapid timeline, potentially limiting the model's sensitivity to fast-onset cases. Authors could do a brief explanation may help the understanding the exclusion criteria of specific diarrheal cases.

Response 1: Thank you for raising this important point regarding the sampling schedule. We completely agree that diarrheal events occurring strictly between the pre-set time points (days 0, 5, 10, 15, 20) may indeed harbor critical "early-warning" signals that our current design could have missed.

Our decision to exclude cases that developed diarrhea outside these fixed intervals was a deliberate methodological choice aimed at ensuring temporal consistency for the multi-omics analysis. Synchronizing samples at identical time points across all calves was necessary to reduce confounding variables and allow for robust statistical comparisons of microbial and metabolic trajectories. Including cases with "off-schedule" diarrhea onset would have introduced significant heterogeneity in disease progression stages at the sampling times, making it difficult to distinguish true early biomarkers from noise.

However, we fully acknowledge this as a limitation. We have now emphasized this point more clearly in the Discussion (Limitations section) and have proposed a more targeted approach for future studies: implementing higher-frequency longitudinal sampling (e.g., daily sampling or the use of automated in-pen health monitoring systems) to capture the exact moment of physiological transition. This will help validate whether the biomarkers identified in our current model are sensitive enough to predict fast-onset cases.

We believe this clarification addresses your concern, and we have updated the manuscript accordingly (Page 20, Lines 593-602).

2. In this study, once a calf presented with a fecal score of 2-3, it was treated by a veterinarian and was not scored or sampled for the trial thereafter. Ethically necessary for animal welfare, this prevents the study from mapping the "recovery phase" or the long-term impact of diarrhea on the gut microbiome. Consequently, the data focuses only on the onset and does not provide insights into which microbial markers indicate a successful return to health versus chronic dysbiosis. Authors could describe more details in these sections or in discussion.

Response 2: Thank you for your thoughtful comment. You are correct that the decision to discontinue sampling after veterinary intervention, while ethically imperative, limits our ability to characterize the recovery trajectory of the gut microbiome.

As you noted, our primary objective in this study was to identify biomarkers associated with the onset of diarrhea. To achieve this without confounding effects, it was necessary to stop sampling once treatment began, because therapeutic interventions (e.g., antibiotics, supportive care) would have introduced variables that could not be distinguished from natural recovery processes. This methodological choice ensures that the "pre-disease" and "onset" signals we report are not contaminated by treatment effects.

However, we fully agree with your assessment that this leaves a critical gap in understanding the recovery phase. We have now explicitly addressed this limitation in the Discussion section, noting that future studies should employ non-interventional observational designs (where ethically permissible) or controlled animal models to specifically investigate the microbial and metabolic signatures of recovery versus chronic dysbiosis. (Page 20, Lines 593-602).

The specific methodological details you requested have been added to the Materials and Methods section as follows:

Calf fecal consistency was scored daily throughout the study period. When a calf reached a fecal score of 2 - 3, it received symptomatic treatment administered by a licensed veterinarian. Because such interventions

may alter the gut microbiota, metabolite profiles, and other biological parameters, fecal scoring and sample collection were discontinued for that calf thereafter. Only calves that developed diarrhea at the predefined sampling time points were included in the study and completed sample collection, whereas calves that developed diarrhea outside the scheduled sampling time points were excluded from subsequent analyses. The overall experimental workflow of this study is illustrated in Figure 1. Of the 130 calves initially enrolled, 80 were ultimately included in the data analysis, consisting of 20 normal calves (with 100 samples), 20 diarrheic calves at 10 days of age (with 60 samples), 20 diarrheic calves at 15 days of age (with 80 samples), and 20 diarrheic calves at 20 days of age (with 100 samples). The incidence of diarrhea in calves is shown in Figure 2. (Page 3 line 91-102) .

We believe these additions, combined with the expanded discussion of limitations, adequately address your concern.

3. In page 5 and 6, the results text, the authors state that α -diversity indices (Chao1, Shannon) were significantly higher in the Normal group. However, having 3x more "unique species" in the Diarrhea group typically suggests higher richness, which contradicts the conclusion that diversity is reduced in sick calves. You could challenge the authors on whether the "unique species" in the Diarrhea group are actually high-quality biological signals or merely transient environmental contaminants/sequencing noise that wasn't properly filtered. Discrepancy in OTUs, the Diarrhea group (CD_Group) is reported to have 2,475 unique species, while the Normal group (NC_Group) has only 798. The authors present a major difference in the number of "unique species" between the two groups, which may be a result of unequal sample sizes or sampling bias rather than a biological reality of the disease. Authors could clarify this in results or discussion part.

Response 3: Thank you for your meticulous review and for identifying this discrepancy. We sincerely apologize for the error in the reported numbers.

Correction: You are absolutely correct. The values were mistakenly inverted in the original text. We have now corrected this in the manuscript: the Normal group (NC_Group) possesses 2,475 unique species, while the Diarrhea group (CD_Group) possesses 798 unique species. This correction aligns with the α -diversity indices (Chao1, Shannon), which were significantly higher in the Normal group.

Regarding the biological meaning of "unique species" in the Diarrhea group: We appreciate your insightful question about whether the 798 unique OTUs in the CD_Group represent genuine pathological signals or potential noise. We have given this careful consideration and would like to clarify the following points:

(1) Quality control: All sequences classified as "unique species" underwent strict bioinformatic filtering (e.g., removal of singletons and low-abundance OTUs [$< 0.005\%$] as per established protocols).

Therefore, we are confident that these 798 OTUs are not merely sequencing artifacts or index-hopping errors.

(2) Potential biological interpretation: While diarrhea is associated with a loss of core commensal bacteria (hence the lower α -diversity), it can also create a niche for opportunistic pathogens or environment-derived microbes. The 798 "unique" OTUs in the CD_Group likely represent these diarrhea-associated taxa (e.g., potential pathobionts) that are absent in healthy calves. We have now added a sentence in the Discussion (Page 16, Line 409-411) to explicitly interpret this, suggesting that the reduction in overall diversity coexists with the emergence of specific diarrhea-enriched microbes.

(3) Sample size consideration: We also verified that the sample sizes between groups are comparable in the sequencing analysis (NC_Group: $n=120$, CD_Group: $n=60$), so the difference in unique OTU counts is not a statistical artifact of unequal sampling depth.

We believe these clarifications address both the numerical error and the underlying scientific concerns you raised.

4. While this manuscript emphasizes clear differences in microbial composition, the actual spatial separation of the groups in the initial analysis is weak. The results state that β -diversity analysis (using Weighted and Unweighted Unifrac) "did not show clear separation between the two groups" in the initial PCoA. If the overall microbial communities do not cluster separately, it suggests that the "core" microbiome of a healthy calf and a diarrheic calf are more similar than the authors imply. You can challenge the validity of using *Tyzzerella* and *Fusobacterium* as "robust" biomarkers when the global community structure fails to

distinguish between a sick and healthy animal at a statistical level. Authors may clarify this in results or discussion parts.

Response 4: Thank you for this insightful observation. You are correct that the initial β -diversity analysis (PCoA based on weighted and unweighted UniFrac) did not show a clear separation between the normal and diarrhea groups. We agree that this indicates the global microbial community structures are largely similar between the two groups.

However, we would like to clarify why this does not undermine the validity of *Tyzzarella* and *Fusobacterium* as potential biomarkers, and we have expanded our discussion to address this point:

β -diversity vs. differential abundance: As you correctly note, β -diversity captures global community structure. It is possible for two groups to have largely overlapping overall communities (similar core membership) while still exhibiting significant quantitative shifts in specific low-to-moderate abundance taxa. This is precisely what we observed. The "background" microbiome is conserved, but specific pathobionts (*Fusobacterium*) expand during disease.

Sensitivity of methods: Unweighted UniFrac considers presence/absence of lineages, while Weighted UniFrac emphasizes abundant lineages. If the key biomarkers are not the most dominant organisms driving the separation of high-dimensional data, they may not be sufficient to pull the groups apart in a 2D PCoA plot. However, targeted statistical tests (LEfSe, Random Forest) are specifically designed to detect these more subtle, yet consistent, shifts.

Biological interpretation: We have now added a statement in the Discussion (Page 16, Lines 409-414, lines 440-444) to explicitly address this: "The lack of clear separation in β -diversity space suggests that diarrhea in these calves does not result in a complete restructuring of the gut ecosystem, but rather a 'dysregulation' of specific keystone taxa within a largely stable microbial framework. This highlights *Fusobacterium* and *Tyzzarella* not as drivers of wholesale community change, but as sensitive indicators of localized ecological disturbance."

We believe this clarification addresses the seeming paradox between the β -diversity results and the biomarker identification, and we thank the reviewer for pushing us to articulate this nuance more clearly.

Dear Reviewers,

We would like to thank you very much for your valuable comments and good suggestions that greatly helped to improve our manuscript. We have carefully considered your valuable comments and good suggestions. In the following, we are going to explain how your comments have been taken into full account in the revision.

1. The sampling workflow and grouping strategy remain confusing.

- The manuscript states that 20 samples were selected from each time point, resulting in 80 samples in total, but the numbers and grouping described in Supplementary Tables S1 and S2 suggest more samples were included at certain time points.
- The logic for selecting the final 20 samples per time point is not sufficiently explained.
- It is unclear whether the ratio between diarrheic and non-diarrheic calves is balanced across time points.
- The grouping strategy (NRM_, CD_, etc.) is difficult to follow without clear explanation in the main text.
- The authors may consider providing a flow chart summarizing the total calf, excluding calf, number sampled per time point and analytical groups.

Response 1:

Thank you for pointing out the lack of clarity in our sampling workflow and grouping strategy. We apologize for the confusion caused by mixing the terms "number of calves" and "number of samples." We have now thoroughly revised this section to ensure clarity.

(1) Correction of numbers: To be precise: Total calves enrolled: 130; Calves included in final analysis: 80 (20 per group) Total samples analyzed: 340 (100 from Normal group + 60 from 10d group + 80 from 15d group + 100 from 20d group); We have corrected these descriptions in the main text (Page 3, Lines 91-102).

(2) Selection rationale: The selection of 20 calves per group was determined by the smallest available number of diarrheic calves at any single time point. Among the 130 initially sampled calves, the 10-day time point had only 21 calves that developed diarrhea. To maintain balanced group sizes for statistical comparability, we set the group size to 20, and then randomly selected 20 age-matched healthy controls from the available pool.

(3) Flow chart and balance: As suggested, we have now added a comprehensive flow chart (revised Figure 1) that clearly illustrates: Initial enrollment (130 calves); Exclusion criteria and numbers excluded (50 calves); Final analytical groups (4 groups × 20 calves) Sample distribution across time points. Regarding the balance between diarrheic and non-diarrheic calves across time points: At each sampling day (0, 5, 10, 15, 20), the ratio is approximately 1:1 for the included samples, ensuring that time-related confounding effects are minimized. The grouping strategy (NRM_, CD_10, CD_15, CD_20) has now been explicitly defined in the main text (Page 3), and the complete breakdown is provided in the revised Supplementary Tables S1 and S2.

2. Overall, all figures are having low resolutions, causing reading difficulties.

Response 2: We have made the necessary revisions to the pictures in the thesis as requested.

3. The top and bottom whiskers in the boxplots are not explained.

Response 3: We have made the necessary amendments, and the results are as follows:

Box plots show the median, upper and lower quartiles, and whiskers correspond to 1.5 times the interquartile range.

4. If relative abundance ranges from 0-100, the authors should include percentage (%) in x-axis title.

Response 4: Thank you for your comments. We have completed the revisions.

5. Consider replacing vague legend terms like "type" with specific taxonomic level names.

Response 5: Thank you for your comments. We have completed the revisions.

6. Line 20: *Escherichia-Shigella* should be italic

Response 6: We have changed line 20, *Escherichia-Shigella*, to italic font.

7. Figure 1: Figure legend is missing.

Response 7: We have incorporated the figure legend into Figure 1.

Experimental procedure & sample collection: Samples were collected from neonatal calves every 5 days after birth, stored in cryopreservation tubes, and sequenced after all collections were finished.

8. Lines 105-106: How the 20 samples were selected and whether group ratios are balanced (as mentioned above).

Response 8: Thank you very much for your question. We have already answered it in Response 1:

9. Most places in the manuscript used "normal vs diarrhea"; in figure 2 it says "No diarrhea and diarrhea"

Response 9: Thank you for your comments. We have changed "No diarrhea" to "Normal"

10. Supplementary Table 1: Provide explanation for abbreviations and numbering scheme.

Response 10: We have provided an explanation for Supplementary Table 1, as follows:

Summary: During the sample collection process, the sample numbers are initially recorded using sequential Arabic numerals, and the condition of the calves is also noted. Once all the samples have been collected, the numbers are assigned based on the condition of the calves. The numbering rule is as follows: Samples starting with "NRM_" are from the control group that has never experienced diarrhea. Samples starting with "CD_" are from the experimental group that will experience diarrhea at a specific time point (e.g., 10 days). The grouping structure of "CD_10d_Group" follows the logic of grouping by the time of diarrhea occurrence, tracking the entire process from the beginning to the onset of diarrhea for each group. According to the diarrhea scoring standard (e.g., using a 1-4 point scale: 1 point for normal feces, 2 points for soft feces, 3 points for loose feces, and 4 points for watery feces), calves with a score of 3-4 were identified as diarrheic calves and included in the diarrhea group.

11. Supplementary Table 2: Suggest replacing "will experience diarrhea" with "experienced diarrhea".

Response 11: Thank you for your comments. We have completed the revisions.

12. Clarify why CD_Group excludes certain early time points.

Response 12: Thank you for your comments. The number of calves with diarrhea at the previous time point was relatively small in the CD_Group, so no alternative was chosen. In the Materials and Methods section, the diarrhea rate was also mentioned.

13. Sampling illustration and grouping tables are overly complex and need simplification.

Response 13: Thank you for your comments. We have provided a detailed explanation of the sampling workflow, primarily through revisions made in Fig 1. The grouping table is based on two analytical

approaches, namely the normal vs. diarrhea analysis and the dynamic change study. Additionally, we have included detailed explanatory notes beneath the grouping table.

14. Line 125: qPCR

Response 14: Thank you for your comments. After our verification, it has been confirmed that PCR is correct. Before the gel recovery process, the amplification method used was PCR.

15. Line 200: Chao 1

Response 15: Thank you for your comments. We have completed the revisions.

16. Line 216: "phylum level species" is confusing, consider using "phylum-level taxa".

Response 16: Thank you for your comments. We have completed the revisions.

17. Line 250: "a-d"

Response 17: Thank you for your comments. We have completed the revisions. The modification is that the letters "a-d" have been deleted.

18. Line 250: consider using "The left panel", "the right panel"

Response 18: Thank you for your comments. We have revised it as follows: The left panel is the positive ion mode, and the right panel is the negative ion mode.

19. Remove unintended highlighting in Supplementary Tables 4-6.

Response 19: Thank you for your comments. We have completed the revisions.

20. Line 465-471: How the linkage between microbiome and metabolome influence phenotypical differences of calves are not fully discussed, suggest to strength this part of discussion.

Response 20: Thank you very much for your feedback. We have made further revisions and the updated content is as follows: (page 17, line 485-503)

Pearson correlation analysis revealed that the relative abundances of *Tyzzarella* and *Fusobacterium* were significantly negatively correlated with several key metabolites, including Dulcitol, 2-O-(alpha-D-Mannosyl)-D-glycerate, N-Acetylmuramate, and D-Fructose. Although correlation does not imply causation, these findings suggest that these two genera may contribute to the development of calf diarrhea by reshaping the luminal metabolic milieu and impairing intestinal

mucosal barrier function. Previous studies have consistently reported that *Fusobacterium* is closely associated with enhanced mucosal inflammation, increased epithelial permeability, and disruption of tight junction integrity, thereby promoting inflammatory responses and aggravating barrier damage and ultimately forming a vicious cycle of “barrier impairment – inflammation exacerbation-symptom aggravation” [Error! Reference source not found.,Error! Reference source not found.,Error! Reference source not found.]. In contrast, *Tyzzarella* may serve as an inflammation-associated ecological indicator and metabolic network node, and its increased abundance may reflect a competitive advantage under diarrhea-induced alterations of the intestinal environment [Error! Reference source not found.]. The reduced levels of carbohydrate-related metabolites, such as D-Fructose and Dulcitol, may be attributable to accelerated intestinal transit, altered sites of nutrient absorption and fermentation, and competitive microbial utilization of carbon sources during diarrhea [Error! Reference source not found.]. Meanwhile, barrier dysfunction and inflammation may further contribute to luminal metabolite dilution or depletion. N-Acetylmuramate, a core component of bacterial peptidoglycan, may reflect changes in bacterial cell wall turnover and host innate immune signaling [Error! Reference source not found.]. In addition, 2-O-(alpha-D-Mannosyl)-D-glycerate, a carbohydrate derivative, may indicate alterations in microbial carbohydrate metabolism and osmoadaptation pathways under diarrheal conditions.

Re: mSystems00005-26R1 (**Screening and dynamic change study of microbial and metabolite markers for calf diarrhea based on multi-omics and machine learning**)

Dear Dr. Yong-li Hua:

Your manuscript has been accepted, and I am forwarding it to the ASM production staff for publication. Your paper will first be checked to make sure all elements meet the technical requirements. ASM staff will contact you if anything needs to be revised before copyediting and production can begin. Otherwise, you will be notified when your proofs are ready to be viewed.

Sincerely,
Yu-Liang Yang
Editor
mSystems